# COMPUTABILITY OF AGENTIC SYSTEMS

## ABSTRACT

This paper introduces the Quest Graph, a formal framework for analyzing the capabilities of agentic systems with finite context. We define abstractions that model common reasoning techniques and establish their computational power: the base Quest Graph is equivalent to an unrestricted Turing machine; the forward-only Finite Quest Decision Process (FQDP), despite its wide use, is only equivalent to a pushdown automaton (context-free); and the Reference-Augmented QDP (RQDP) regains Turing completeness only when stateful queries are allowed. Since computability affects efficiency, we then analyze the theoretical efficiency of each model by simulating task dependencies in computation graphs. We show that this computational hierarchy translates to concrete performance trade-offs: reference-augmented (Turing-complete) systems can be exponentially more efficient at simulating complex graphs than their non-augmented (context-free) counterparts. This work provides a formal methodology for classifying and understanding the fundamental capabilities of agentic systems.

## 1 INTRODUCTION

Agentic AI is an emerging term for systems capable of autonomous action (Durante et al. (2024); Acharya et al. (2025)). Operating with minimal supervision, these systems maintain their own goals and strategically utilize available resources to achieve them. Language models (LMs) lie at the core of these systems, serving as the primary decision-making module that performs complex reasoning and planning by processing natural language.

Several strategies exist for improving agentic system performance. The most direct, increasing an LM's parameter size, adds computational demand to train and execute. Another approach involves expanding the reasoning steps to cover multiple perspectives (Wei et al. (2022); Wang et al. (2022); Liu et al. (2023); Shinn et al. (2023); Zhao et al. (2023); Wang et al. (2024)). These techniques require increasing amounts of context to handle their meta-reasoning. On the contrary, larger context is not always required for better performance (Shi et al. (2023); Liu et al. (2024); Jiang et al. (2024)), as it can introduce irrelevant information that distracts the model. In fact, another class of techniques enhances performance by decomposing complex problems into simpler, hierarchical sub-tasks, which can be solved with a smaller context size (Zhou et al. (2022); Wang et al. (2023); Chen et al. (2024); Yao et al. (2024); Besta et al. (2024); Ma et al. (2025)). These techniques call into question the attempt to improve performance solely by increasing context size, prompting a more fundamental analysis.

In this work, we pose the question of how capable a system can be given a fixed context size. This is related to classical computability theory, which studies what can be computed given finite computational resources (Hopcroft et al. (2001)). It turns out that different levels of computability can affect not only what can be computed, but also how efficiently it can be computed in some settings.

Imagine a collection of repeatable processes $A$, $B$, and $C$, where $B$ is asynchronous (Syme et al. (2011)), and $C$ depends on $B$ and $B$ depends on $A$. A system limited to regular expressions can only process $A$, $B$, and $C$ in a fixed sequence, recognizing a pattern like $(ABC)+$. Whereas a system with context-sensitive computability can process the $A^n B^n C^n$ pattern. The latter enjoys greater efficiency by pipelining the asynchronous $B$ processes before proceeding to $C$, thereby ensuring minimal waiting time. There could also be many other scenarios where different computability levels lead to different efficiency trade-offs.

Instead of relying on illustrative scenarios, we perform our analysis more formally using dependency graphs. The question we ask is: given a preferred pattern of execution, how effectively can agentic systems with different computational capabilities simulate it? To answer this, we propose a formal framework called the *Quest Graph*, a name inspired by the goal-oriented nature of agentic systems. This framework classifies agentic systems by demonstrating that their common inference patterns correspond to distinct levels within the formal language hierarchy (Chomsky (1956); Hopcroft et al. (2001)). We then analyze the efficiency with which each system variant can simulate computation graphs, a powerful tool for describing complex computational relationships (Abadi et al. (2016)). We believe this investigation is a foundational step toward a formal understanding of the computability and capabilities of agentic systems.

| Languages | Automata | Agentic Systems | Reasoning Features | CG Complexity |
|---|---|---|---|---|
| Unrestricted | TM | Quest Graph | Graph traversal | $O(N^2)$ |
| | | RQDP, NRQDP | Knowledge retrieval | $O(N^2 \log N)$ |
| Context-free (CFL) | PDA | NFQDP | Search | $O(2^N)$ |
| Deterministic CFL | DPDA | FQDP | Hierarchy | $O(2^N)$ |
| Regular | FSA | LM | Autoregression | n/a |

Table 1: Summary of main results: Alignment of agentic architectures with the formal language hierarchy. The table maps each system variant to its corresponding automaton class and primary reasoning capability. The final column reports the time complexity required to simulate a computation graph (CG) with $N$ nodes; entries marked "n/a" indicate models incapable of simulating the graph under constant response size constraints.

## 2 AUTOREGRESSIVE LANGUAGE MODELS WITH FINITE CONTEXT

To establish a baseline for agentic computability, we first consider a standard Language Model (LM) operating on a finite context window— a fixed-size memory buffer containing the sequence of recent tokens the model can access. We define this LM as a model that progressively generates the next token based on the current contents of this window. At each computational step, the context is updated in two ways: the model's own output is autoregressively appended, and any new external input is also added.

This abstraction allows us to analyze the LM's computational capabilities without delving into specific architectures or training methods which may affect the model's computability. For example, recurrent neural networks and long short-term memory architectures (Hochreiter & Schmidhuber (1997)) are sufficient for regular language recognition, while transformer-based models have been shown to be weaker (Hahn (2020)).

For this analysis, we assume the model's output is deterministic, equivalent to setting its sampling temperature to zero in many energy based implementations (Vaswani et al. (2017)).

Despite their impressive performance, LMs with finite context windows are fundamentally limited. Given that the vocabulary of tokens is finite, it can be formally shown that:

**Theorem 1.** *An LM with a finite context is computationally equivalent to a finite state machine (FSM).*

This theoretical equivalence has been shown for autoregressive mechanisms by Delétang et al. (2022). It highlights a crucial distinction: finite-context LMs are equivalent to FSMs (Hopcroft et al. (2001)), whereas their unbounded-context counterparts are Turing complete (Pérez et al. (2021)). Nevertheless, we provide a proof in Appendix E to formally establish this baseline and to make the paper self-contained.

The standard model of an LM treats the decision-making module and the context buffer as a single, monolithic entity. We propose decoupling these components, presenting the reasoning module as a pure function that operates on an external memory structure. This separation offers both practical and theoretical advantages. On a practical level, it aligns with modern, stateless architectures recognized for their scalability and testability (Fielding (2000)). More importantly, this approach allows us to

define a hierarchy of agentic systems with increasing computational power, effectively scaling the complexity hierarchy.

# 3 QUEST GRAPHS AND AGENT FUNCTIONS

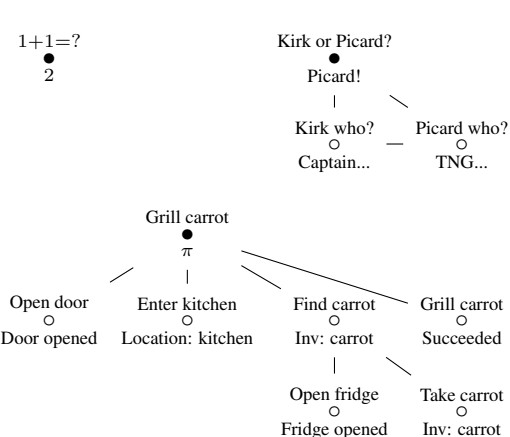

Figure 1: Illustrative Quest Graphs. Top Left: A single-node graph. Top Right: A graph representing a multi-hop comparison task. Bottom: A 2-level hierarchical graph rollout for an RL problem. The focus node is the one with a filled circle. $\pi$ represents a parent mark (defined in Section 4).

The concept of a "Quest Graph" is inspired by working memory models that describe components supporting human cognitive functions (Baddeley (1992)). Working memory provides temporary storage for information accessible to higher cognitive processes like decision-making. Adopting a thematic naming convention inspired by quests, we refer to the decision-making module as an "agent". The interaction between these two components constitutes our proposed framework.

## 3.1 DEFINITIONS AND MECHANICS

Let a **Quest Graph** be a dynamic, undirected graph $\mathcal{Q} = (V, E)$. The set of nodes, $V$, is composed of elements from a finite set of node configurations, $\mathcal{V}$. The set of edges, $E$, connects pairs of nodes. The Quest Graph also maintains a reference to a specific **focus node**, $v_f \in V$.

Each node configuration $v \in \mathcal{V}$ is a tuple $(g, r)$ consisting of two components: a static **goal**, $g \in \mathcal{G}$, which is set upon the node's creation and remains unchanged, and a dynamic **response**, $r \in \mathcal{R}$, which the agent can modify during the computation. The response set, $\mathcal{R}$, may include a special empty symbol, $\varepsilon$, to signify an **incomplete** state. Both the set of all possible goals, $\mathcal{G}$, and all possible responses, $\mathcal{R}$, are finite.

The agent operates on the Quest Graph by reading data from the focus node and its neighbors to infer its next action, using the graph as its exclusive working memory.

Let $C \in \mathbb{Z}^+$ be a positive integer that defines the maximum number of neighbors the agent can consider at one time. This parameter $C$ does not restrict a node's true degree in the graph but rather limits the agent's simultaneous processing capacity, analogous to a cognitive load limit. Let $\mathcal{V}^{\leq C} = \bigcup_{k=0}^{C} \mathcal{V}^{1+k}$ be a finite set that includes all configurations of a focus node and an ordered set of up to $C$ of its neighbors. Let $\mathcal{E}$ represent a set of edge pointers referring to one of the neighbors of, or a self-loop on, the focus node. Let $\mathcal{E}^{\leq C} = \bigcup_{k=0}^{C} \mathcal{E}^{1+k}$ be a finite set that includes all configurations of edge pointers to the focus node and up to $C$ of its neighbors.

Next, we define the **agent function**, $\chi$, which takes the current **local context** of the Quest Graph—an element from $\mathcal{V}^{\leq C}$ representing the focus node and its neighbors—and outputs an action. There are three possible actions: 1. **Discover node**, which creates a new node with a specified goal and attaches it to selected nodes from the current context. The output is an element of $\mathcal{V} \times \mathcal{E}^{\leq C}$. 2. **Respond then move focus**, which updates the response of the focus node and shifts the focus to one of its neighbors. The output is an element of $\mathcal{R} \times \mathcal{E}$. 3. **Stop execution**, which terminates the computation and returns no parameters.

Through these actions, the Quest Graph evolves in a manner analogous to a player progressing through quests in a role-playing game. An example of this evolution is depicted in Figure 2.

A key theoretical question thus arises from these mechanics: can an agent, limited by a finite context capacity $C$, achieve universal computation within this framework?

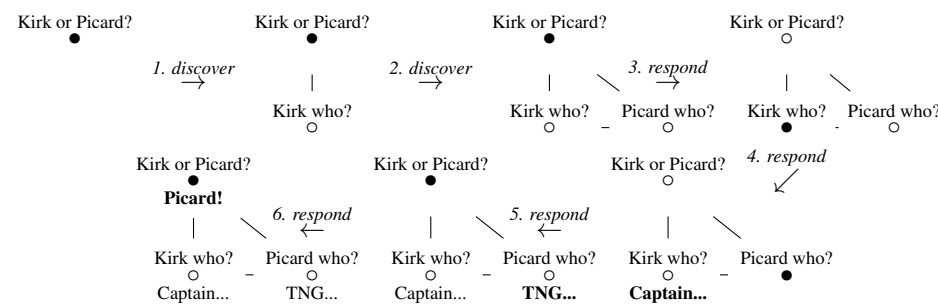

Figure 2: Evolution of a Quest Graph for a multi-hop question answering task. Steps 1-2: "Discover node" actions generate new nodes for sub-quests. Step 3: A "Respond then move focus" action returns empty update to the main node but shifts focus to a sub-quest. Steps 4-6: Subsequent "Respond then move focus" actions provide answers to the sub-quests and, finally, to the main quest node.

## 3.2 TURING COMPLETENESS

To establish the computational universality of our framework, we demonstrate that the Quest Graph system is Turing complete by constructing an agent function that simulates a standard Turing machine (TM) (Turing (1936); Hopcroft et al. (2001)) using the Quest Graph as its operational environment. Specifically, we consider a TM with an infinitely long tape, where the head can move either left (L) or right (R) at each step.

**Theorem 2.** *An agent operating on a Quest Graph with a finite context size is Turing complete.*

Our agent function is stateless and relies entirely on the Quest Graph for memory. Therefore, our construction is analogous to an unconventional TM variant. In this model, the state is not held in a separate internal register but is encoded directly on the tape, where the head can also see adjacent cells to determine its next action. While similar unconventional models exist in the literature (Arulanandham (2007); Cantone & Cristofaro (2016)), we provide a proof sketch here tailored to our framework. A more detailed proof is provided in Appendix A.

*Proof Sketch.* We represent the TM's tape as a linear Quest Graph where each node corresponds to a tape cell. Each node's response stores the necessary TM configuration: a tape symbol, a state symbol, and a direction symbol (L or R). The simulation proceeds in steps where the agent receives the local context, takes the TM's current configuration from it, writes the next configuration by responding to the focus node, and then shifts the focus as dictated by the TM's transition function. This mechanism ensures that the current tape symbol is always stored in the focus node's response, while the TM's current state is stored in the response of the previous node, which is one of the focus node's neighbors.

Critically, the agent can unambiguously identify this previous node by inspecting the direction symbols. Because a TM cannot skip cells and the tape is acyclic, the agent uses a simple rule: the previous cell is the neighbor whose direction symbol is the opposite of the focus node's. □

While the general Quest Graph framework is Turing complete, its unrestricted nature presents practical challenges. The ability to rewrite any node's response creates a combinatorial explosion in the state space, which makes learning an effective agent policy difficult. Furthermore, modifying an agent's history is often undesirable in applications where a clear and immutable record of action causality must be preserved for learning.

This leads us to model a more constrained, yet common, pattern of agentic reasoning defined by two rules: agents can only discover nodes in a forward direction, and they cannot rewrite the responses of past nodes. These restrictions naturally encourage a hierarchical reasoning approach. To solve a complex problem, the agent must decompose it into simpler sub-tasks, which we term ***sub-quests***. This process expands a reasoning tree, where the agent adds a new child node (a sub-quest) to its current focus and uses the context from the entire branch to inform its next action. This hierarchical pattern is prevalent in many agentic systems (Ahn et al. (2022); Yao et al. (2023); Schick et al. (2023);

Song et al. (2023); Tang et al. (2024)) and is a cornerstone of hierarchical reinforcement learning (HRL) (Barto & Mahadevan (2003); Sutton et al. (1999)).

# 4 QUEST DECISION PROCESSES

We introduce the ***Quest Decision Process*** (QDP), a more constrained version of the Quest Graph framework designed specifically to model hierarchical reasoning. The QDP operates under four key constraints: 1. it imposes a strict structural rule where new nodes are linked exclusively as children of the current focus node; 2. the discover action is separated into two distinct types. A ***discover input*** action creates a new node to store external information, with its goal as the query and its response as the information received. A ***discover sub-quest*** action creates a new node with a specified goal and an empty response, after which the agent marks the current focus node as a parent and immediately shifts focus to this new child; 3. the agent can only "respond then move" back to a parent node after completing its current quest; and 4. the agent can only stop the execution when the focus node is the root node. The bottommost diagram in Figure 1 illustrates a typical QDP rollout.

Formally, a QDP is a variant of the partially observed Markov decision process (POMDP), whose defining tuple includes a state space ($\mathcal{V}$), a hierarchical action space ($\mathcal{G}$), an agent function ($\chi$), an observation space ($\Omega_{\text{qdp}}$), an observation function ($\mathcal{O}_{\text{qdp}}$). The $\mathcal{V}$ and $\mathcal{G}$ correspond to the sets of nodes and goals in the Quest Graph, respectively. Specifically, $\mathcal{G}$ is a union of the original POMDP's action space and a set of discoverable, high-level sub-quest goals. Similarly, the node response set, which we denote as $\mathcal{R}$, is a union of the original POMDP's observation space, a set of sub-quest outcomes, and a special configuration symbol $\pi$ indicating the quest is a parent node of a sub-quest. The composition of this result set is a flexible design choice; for instance, it could include discrete outcomes like $\{\text{succeeded}, \text{failed}, \text{truncated}, \text{unfulfilled}\}$ or other forms of summary information. Together, the goals and responses of the active Quest Graph nodes form the context for the agent's decisions.

While the general QDP is Turing complete, mirroring the capabilities of infinite-context LMs (Pérez et al. (2021)), our analysis focuses on a more constrained model: the ***Finite Quest Decision Process*** (FQDP). The FQDP restricts each node to a finite number of children. Once this child limit is reached, the agent can only perform actions that do not create additional children for the focus node. These constraints fundamentally reduce the model's computational power.

**Theorem 3.** *A Finite Quest Decision Process (FQDP) is equivalent to a deterministic pushdown automaton (DPDA).*

The proof, detailed in Appendix B, involves a bisimulation between the FQDP and a deterministic pushdown automaton (Hopcroft et al. (2001)). This equivalence is intuitive: the FQDP's hierarchical, last-in-first-out process of managing sub-quests directly mirrors how a stack handles function calls in modern programming languages.

## 4.1 NON-DETERMINISTIC FINITE QUEST DECISION PROCESSES

The sub-quest mechanism in the FQDP encourages a deterministic, depth-first exploration of the reasoning tree. The primary benefit of this determinism is that it simplifies policy learning, as the resulting execution trace is unambiguous. However, another commonly adopted pattern is breadth-first exploration, where an agent creates multiple sub-quests or solutions in parallel and uses external mechanisms to evaluate and select among them (Yao et al. (2024); Hao et al. (2023); Wang et al. (2022)).

We call this variant the ***Non-deterministic Finite Quest Decision Process*** (NFQDP). The NFQDP relaxes the FQDP's strict sub-quest discovery process by further splitting it into two separate actions: 1. the discover sub-quest action now creates a new child node without immediately shifting focus to it, and 2. a new pursuit action allows the agent to shift focus to any of the current focus node's incomplete neighbors.

As the name suggests, this model is computationally equivalent to a non-deterministic pushdown automaton (PDA) (Hopcroft et al. (2001)). The separation of the discover and pursue actions allows the agent to traverse a ***pre-built*** reasoning tree. This is necessary for cases where a single quest

configuration might lead to different reasoning paths, a distinction that is indeterminable by the agent until the sub-quest is actively pursued. A full proof is provided in Appendix C.

Systems with a finite context that process information hierarchically fall into this category. For example, ReAct is an FQDP system, as it interleaves reasoning and tool use in a depth-first manner (Yao et al. (2023)). HRL agents also fit this model, where a high-level policy sets sub-goals for a low-level policy to achieve in a nested fashion (Sutton et al. (1999); Barto & Mahadevan (2003)). On the other hand, systems like Tree of Thoughts (Yao et al. (2024)) and Reasoning via Planning (Hao et al. (2023)), which explore alternative nested paths via external search mechanisms, can be modeled as NFQDPs.

These results have a significant implication: any agentic system that operates under the constraints of either an FQDP or an NFQDP is not Turing complete. Instead, its computational power is bounded, equivalent to that of a machine capable of recognizing context-free languages. This reduction in computational power highlights the need for a model that is not only more expressive than a PDA but that can also preserve an immutable history of actions, all while operating within a finite context window.

## 5   REFERENCE-AUGMENTED FINITE QUEST DECISION PROCESSES

Inspired by the retrieval mechanism for knowledge access in Retrieval-Augmented Generation (RAG) systems (Lewis et al. (2020)), we propose the **Reference-Augmented Finite Quest Decision Process** (RQDP). Typical RAG systems query an external knowledge base to enhance generative performance. Our mechanism, however, uses retrieval to access the agent's own past experiences—nodes that are inaccessible to the local context. This provides access to an unbounded history while also ensuring the agent's rollout remains immutable, which is a desirable property for policy learning.

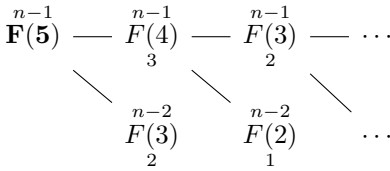

Figure 3: An RQDP simulating the recursive computation of a Fibonacci number, $F(n) = F(n-1) + F(n-2)$. Top: The execution trace (rollout), where the agent explores the dependency graph depth-first, recursively computing the $F(n-1)$ term while retrieving the pre-computed value for the $F(n-2)$ term. For termination, we assume an omitted input is discovered at each $F(n-1)$ node to count down to the base case. Bottom: The corresponding reference graph, where edges are defined by the reference-generating function, $\tau$. The label in each node (e.g., $F(5)$) represents its reference; the corresponding node in the reference graph stores the cached response.

To enable the retrieval mechanism, the system requires a way to distinguish nodes. Each node is associated with a tag called a **reference**, drawn from a non-finite set $\mathcal{T}$. This set must be non-finite to accommodate the potentially unbounded history of the Quest Graph. If the reference set were finite, an agent could theoretically encode the entire configuration of references into a single finite response, which would not fundamentally increase the computational power beyond that of an FQDP. This limitation is inherent to the stateless retrieval mechanisms in common RAG systems, where retrieval relies solely on a finite query derived from the current context (Gao et al. (2023); Peng et al. (2024)).

When a new node is created via any action, the Quest Graph assigns it a reference using a deterministic **reference-generating** function, $\tau : (\xi, g) \mapsto \xi'$. This function maps the focus node's reference ($\xi \in \mathcal{T}$) and the new node's goal ($g \in \mathcal{G}$) to a new reference $\xi' \in \mathcal{T}$.

The RQDP introduces a **retrieve** action, defined as $\mathcal{V}^{\leq C} \to \mathcal{G}$. When the agent selects this action, it outputs a goal. The Quest Graph then triggers the creation of a new node, analogous to the discover action. But instead of immediately initializing the new node's response to be empty, the Quest Graph tries to search the history for a matching node to copy its response. A historical node is considered a match if: 1. its reference matches that of the new node according to a matching operator, $\mathcal{T} \times \mathcal{T} \to \{\text{true}, \text{false}\}$; and 2. it is

the most recently updated node among all matches in the history. If no match is found, the retrieval node's response is set to the empty symbol, $\varepsilon$.

From the construction of the reference-generating function, the reference mechanism must maintain state to ensure sufficient capacity for distinguishing references over an unbounded history. Crucially, this design does not violate the agent function's finite and stateless assumption. The unbounded reference tags and the stateful retrieval process are managed by the Quest Graph, while the local context presented to the agent function remains finite, effectively isolating the agent from this underlying complexity.

The reference mechanism can be realized by implementing a separate reference graph that stores the current response for each unique reference. In this implementation, the reference graph maintains its own head that tracks the reference associated with the current focus node and moves as the focus changes. New nodes are added to this graph as they are created, with their connections defined by the $\tau$ function. When the agent responds to a node in the main Quest Graph, the system updates the response of the corresponding node in the reference graph. Figure 3 illustrates an RQDP rollout, where the bottom diagram depicts the corresponding reference graph.

### 5.1 COMPUTABILITY OF RQDP

The construction of the RQDP is conceptually similar to that of auxiliary PDAs and stack automata. However, these models are typically constrained to bounded auxiliary memory, placing them in the LOGCFL and PSPACE complexity classes, respectively (Sudborough (1978); Ginsburg et al. (1967); Hopcroft & Ullman (1967)). Our model differs in two key aspects: its reference mechanism provides a more flexible access pattern, and it allows for an unbounded history. These features are what elevate the RQDP to Turing completeness.

**Theorem 4.** *A Reference-Augmented Finite Quest Decision Process (RQDP) is Turing complete.*

*Proof Sketch.* We show that an RQDP can simulate a standard Turing machine (TM) by using its reference mechanism to represent the TM's tape positions. Because a standard TM has a one-way infinite tape, this simulation only requires a reference-generating function capable of incrementing the reference, which corresponds to the rightward movement of the tape head. A full proof is provided in Appendix D.

The simulation represents the TM's tape in two parts. The portion of the tape to the left of the head is maintained by the RQDP's stack-like execution, where pursuing a sub-quest corresponds to a "push." During this push, the agent overwrites the parent's response with a parent mark ($\pi$), which is why the tape symbol for this portion is stored directly in the state of the parent node. The portion of the tape to the right of the head is managed by the reference mechanism, where each reference corresponds to a unique tape cell, and its symbol is retrieved when needed.

The agent's actions are as follows: 1. To move right, the agent discovers a new child node, which is assigned an incremented reference ($\xi + 1$) by the $\tau$ function, extending the tape to the right. To read the symbol at this new position, it uses a retrieve action. 2. To move left, the agent completes the current node and moves the focus to its parent, effectively "popping" the stack. The tape symbol for this newly revealed position is read directly from the parent node's state. To allow for a subsequent rightward move from this "revisited" parent, a "replica" node is used to prevent the formation of a cycle while preserving the finite context constraint. $\qquad\square$

Additionally, we define a ***Non-deterministic Reference-Augmented Finite Quest Decision Process*** (NRQDP) by augmenting the NFQDP with the same reference mechanism used in the RQDP. Since the NRQDP contains all the capabilities of the deterministic RQDP, it is also Turing complete.

The reference mechanism enables both the RQDP and NRQDP to surpass the computational bounds of context-free languages while preserving execution traces for learning. It appears that such a mechanism is a necessary component for a hierarchical agent to achieve at least a context-sensitive level of complexity.

Having established a formal hierarchy of agentic automata corresponding to distinct levels in the formal language hierarchy, we now shift our focus from pure computability to theoretical efficiency.

We investigate this by asking a key question: how efficiently, in terms of computational steps, can each model in our hierarchy simulate a general computation graph?

# 6 SIMULATING COMPUTATION GRAPHS

A computation graph is a directed acyclic graph (DAG) where nodes represent operations and edges define the dependencies between them. The graph's structure dictates the order of execution: a node can only be computed after all of its predecessors (nodes with edges pointing to it) have been computed.

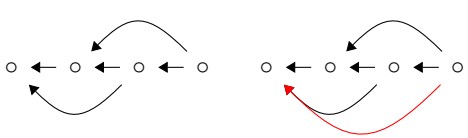

Figure 4: Computation graph (left) and its max dependency version (right) with equal number of nodes.

To analyze the worst-case efficiency of the Quest Graph and its variants, we introduce a standardized benchmark: the ***Maximum-Dependency Computation Graph*** (MCG). An MCG is a maximally connected DAG where, for a given ordering of nodes, each node depends on all preceding nodes. We establish that any computation graph is a subgraph of an MCG (see Appendix F for a proof).

This approach allows for a consistent analysis independent of any specific graph configuration.

An MCG with $N$ nodes has exactly $\frac{N(N-1)}{2}$ edges, providing a clear basis for comparing the simulation efficiency of our agentic automata. Figure 4 shows an example of a computation graph and its MCG counterpart.

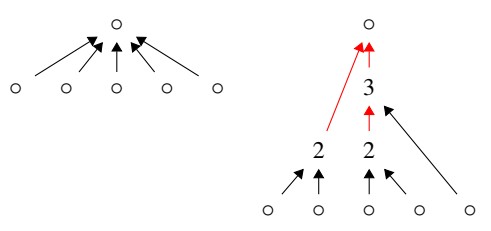

Figure 5: A node of degree five (left) and its k-ary tree of proxy nodes for $C = 2$ (right). The numbers represent the proxy nodes and their potential increase in the response size. The red edges are the added proxy edges.

A practical issue arises when an MCG's maximum in-degree (the number of dependencies for a node) exceeds the maximum context size, $C$. Since $C$ can be much smaller than the total number of nodes $N$, an agent cannot process all dependencies of a highly connected node at once. To resolve this, we convert the graph into a structure compatible with the agent's limited context. Any node with an in-degree greater than $C$ is decomposed into a k-ary tree of ***proxy nodes***, where each new node in the k-ary tree has at most $C$ dependencies (Figure 5). We call the resulting structure a ***Bounded Maximum-Dependency Computation Graph*** (BMCG).

**Lemma 5.** *Given a Maximum-Dependency Computation Graph (MCG) with $N$ nodes, there exists an equivalent Bounded Maximum-Dependency Computation Graph (BMCG) composed of $O(N^2)$ nodes and $O(N^2)$ edges.*

*Proof Sketch.* The quadratic complexity arises from summing the conversion cost across all nodes. For any single node, the number of required proxy nodes is linear in its "excess degree"—the amount by which its in-degree exceeds the context size $C$. In an MCG with $N$ nodes, the in-degrees of the nodes increase linearly, forming an arithmetic progression. The total number of added nodes is therefore the sum of a linear sequence of these excess degrees, which results in a quadratic total, $O(N^2)$. Since each added proxy node also corresponds to a constant number of new edges, the total number of edges is also quadratic. A full numerical analysis is provided in Appendix F. □

Decomposing a node has a potential cost, as proxy nodes must carry intermediate responses. The k-ary tree structure is designed to ensure this information flow is well-balanced among the new nodes. However, the required response size depends entirely on the nature of the original computation.

In the most extreme case, the response size can grow significantly because one could trivially define a single node whose computation requires knowledge of the entire preceding graph. For a non-

compressible function, a proxy node might simply concatenate the responses from its children, leading to a linear increase in response size. In contrast, many common operations are highly compressible. A summation requires only logarithmic space for a running sum, while operations like boolean logic or finding a maximum value require only constant space. Therefore, the response size increased by this transformation has a lower bound of $\Omega(1)$.

## 6.1 COMPLEXITY ANALYSIS

We now analyze the time complexity required for each Quest Graph variant to simulate a computation graph. To do so, we first establish the computational cost of the agent function. Since the agent operates on a finite local context, we assume it computes its output in constant time, $O(1)$.

For the unrestricted Quest Graph, the BMCG can be pre-built within the Quest Graph, with the root node as the focus node. To perform the computation, the agent can traverse the graph in a depth-first manner until it finds a node whose dependencies have all been computed. The time complexity for traversing the graph is bounded by the number of nodes, because for each non-leaf node, the agent must visit and then backtrack from it.

**Corollary 6.** *A Quest Graph can simulate a BMCG with $O(N^2)$ operations.*

Unlike the unrestricted Quest Graph, an NRQDP cannot directly instantiate a BMCG due to its single-parent constraint. Instead, the NRQDP must embed the BMCG's exploration tree within its own quest graph, a process that would typically entail a number of nodes exponential in the size of the underlying MCG. However, the reference mechanism allows the NRQDP to share a node's response across different branches of this exploration tree. This is achieved by assigning a unique reference to each unique node from the initial MCG, which substantially reduces the number of nodes that must be computed.

The deterministic RQDP operates differently, as it cannot navigate through a pre-built exploration tree due to its more restricted action set. Instead, a node's dependencies must be provided to the agent through discover input actions. This forces the agent to discover the graph's structure dynamically during execution while using its reference mechanism to retrieve previously computed nodes.

For both the NRQDP and RQDP, the simulation proceeds as a memoized, depth-first traversal. To compute any given node, the agent retrieves each of its dependencies using their references. If a retrieved dependency has an empty response, indicating it has not yet been computed, the agent recursively invokes the simulation on that dependency first. To ensure all dependencies are resolved, this process requires the agent to check every edge of the underlying MCG. Since the total complexity is the product of the number of edges and the cost of each retrieve action, we can conclude that:

**Corollary 7.** *An NRQDP and RQDP can simulate a BMCG with $O(N^2 \log N)$ operations.*

Here, we assume the retrieve action has a logarithmic time complexity relative to the number of active references, $O(\log N)$. This reflects the inherent cost of finding a matching reference within the unbounded set $\mathcal{T}$. In practice, the reference mechanism can be implemented with a key-value store, which typically entails a logarithmic access time (Kleppmann (2017)). Alternatively, if implemented via a separate reference graph (as described earlier), the logarithmic cost is incurred not during reference generation, but during the respond then move action when searching for the correct parent node's reference to return to. Both implementation strategies yield the same asymptotic complexity.

An NFQDP and FQDP traverse the graph dynamically, similar to their reference-augmented counterparts, but they lack the ability to refer to past results. Without this capability, the agent must re-compute shared dependencies each time they are needed. This re-computation is necessary for the constant response size lower bound, which prevents them from storing the results of large subtrees directly in the response field.

**Corollary 8.** *An NFQDP and FQDP can simulate a BMCG with $O(2^N)$ operations.*

*Proof.* The cost of computing a dependency subtree has two components: the computation within the node's immediate proxy tree and the recursive computation of its underlying dependencies. The cost of the proxy tree is proportional to its number of proxy nodes, which in turn is proportional to the difference between the node's in-degree and the context size $C$.

Let $S(N)$ be the number of operations to compute a node with $N-1$ dependencies. We can establish both a lower and an upper bound for this cost with a recurrence relation. The lower bound is given by $S^-(N) = 1 + \sum_{i=1}^{N-1} S^-(i)$, which solves to $S^-(N) = 2^{N-1}$. The upper bound is given by $S^+(N) = N + \sum_{i=1}^{N-1} S^+(i)$, which solves to $S^+(N) = 2^N - 1$. $S^-(1) = S^+(1) = 1$.

Since both the lower and upper bounds are $O(2^N)$, the size of the subtree to be re-computed is bounded by $O(2^N)$. $\qquad\square$

Finally, a standard LM is fundamentally more limited. Unlike the Quest Graph variants that provide structured, graph-like memory, an LM operates on a simple, linear context window of a fixed size, $C$. To simulate a BMCG, the LM must individually store previously computed intermediate results within this single window. As the dependency size increases, the number of intermediate results that must be stored scales with the height of the k-ary proxy tree, a logarithmic factor of $N$, eventually exceeding $C$.

**Corollary 9.** *A standard LM cannot simulate an arbitrary BMCG when restricted to a constant response size lower bound.*

This concludes our analysis of the Quest Graph and its variants in the context of simulating computation graphs.

## 7 CONCLUSION

In this work, we introduced the Quest Graph, a theoretical framework for analyzing the computational capabilities of agentic systems with finite context windows. We demonstrated that architectural variants of this framework, designed to model common reasoning patterns, correspond to distinct levels of the formal language hierarchy: 1. The unrestricted Quest Graph for graph traversal is Turing complete. 2. The Finite Quest Decision Process (FQDP) and Non-deterministic FQDP (NFQDP) for hierarchical reasoning are equivalent to deterministic and non-deterministic pushdown automata, respectively. 3. The Reference-Augmented FQDP (RQDP) and Non-deterministic RQDP (NRQDP) are also Turing complete, provided they are equipped with a reference mechanism that maintains state. This establishes a clear link between an agent's design and its fundamental expressive power.

Furthermore, we analyzed the theoretical efficiency of these models when simulating general computation graphs and established the following time complexities: 1. The unrestricted Quest Graph can simulate a bounded maximum-dependency computation graph (BMCG) in $O(N^2)$ operations. 2. The RQDP and NRQDP can simulate a BMCG in $O(N^2 \log N)$ operations. 3. The FQDP and NFQDP can simulate a BMCG in $O(2^N)$ operations. 4. A standard LM cannot simulate an arbitrary BMCG when restricted to a constant response size. A summary of our findings is presented in Table 1.

## 8 DECLARATION OF LM USAGE

In the creation of this manuscript, we utilized AI language models for assistance *only with final text polishing and literature searches*. This decision was guided by our conviction that the process of intellectual discovery is itself a rewarding and essential part of the scientific endeavor. We affirm that all core intellectual contributions, conceptualization, and the ultimate responsibility for this work rest entirely with the human authors.

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

## A COMPUTABILITY OF QUEST GRAPH

**Definition 10.** *A **Quest Graph** is a dynamic, undirected graph constructed from a finite set of components. These components are:*

- *A finite set of all possible goals, $\mathcal{G}$.*

- *A finite set of all possible responses, $\mathcal{R}$. The response set may include a special empty symbol, $\varepsilon$, to signify an **incomplete** state.*

*A node configuration is a tuple $(g, r)$, where $g \in \mathcal{G}$ and $r \in \mathcal{R}$. Both $\mathcal{G}$ and $\mathcal{R}$ are finite sets, ensuring that the set of all possible node configurations, $\mathcal{V} = \mathcal{G} \times \mathcal{R}$, is also finite.*

*A specific instance of a Quest Graph is a tuple $\mathcal{Q} = (V, E)$, where:*

- *$V$ is the set of nodes currently instantiated in the graph. Each node $v \in V$ has a configuration from the set $\mathcal{V}$.*

- *$E$ is the set of undirected edges between the nodes in $V$.*

*The graph maintains a reference to a specific **focus node**, $v_f \in V$.*

*A **local context** consists of the focus node and its neighbors up to a fixed capacity $C \in \mathbb{Z}^+$. The set of all possible local contexts is denoted by $\mathcal{V}^{\leq C} = \bigcup_{k=0}^{C} \mathcal{V}^{1+k}$. Let $\mathcal{E}$ represent a set of edge pointers referring to one of the neighbors of, or a self-loop on, the focus node. The set of all possible edge pointers is denoted by $\mathcal{E}^{\leq C} = \bigcup_{k=0}^{C} \mathcal{E}^{1+k}$.*

*The agent function, $\chi$, takes a local context and performs one of three possible actions:*

- *Discover node: Returns a new node and a set of edge pointers that point toward neighbors to be connected to the new node. This output is an element of $\mathcal{V} \times \mathcal{E}^{\leq C}$, where $\mathcal{E}^{\leq C}$ is the set of edge pointers.*

- *Respond and move focus: Returns a response for the current node and an edge pointer indicating the next focus node. This output is an element of $\mathcal{R} \times \mathcal{E}$.*

- *Stop: Returns nothing and halts the computation.*

Let a Turing machine be defined by the 7-tuple $(S_{tm}, \Gamma_{tm}, \Sigma_{tm}, \delta_{tm}, s_{tm}^0, b_{tm}, F_{tm})$. For our simulation, the key components are the finite set of states ($S_{tm}$), the tape alphabet ($\Gamma_{tm}$), the initial state ($s_{tm}^0 \in S_{tm}$), the transition function $\delta_{tm} : S_{tm} \times \Gamma_{tm} \to S_{tm} \times \Gamma_{tm} \times \{L, R\}$, and the set of accepting states $F_{tm} \subseteq S_{tm}$. The remaining components—the input alphabet $\Sigma_{tm} \subseteq \Gamma_{tm}$, and the blank symbol $b_{tm} \in \Gamma_{tm} \setminus \Sigma_{tm}$—are part of the standard definition but are not central to the mechanics of the simulation.

**Theorem 11.** *There exists an agent function with a context capacity of $C = 2$ that, by operating on a Quest Graph, can simulate an arbitrary Turing machine.*

*Proof.* We represent the Turing machine's tape as a linear Quest Graph where each node corresponds to a tape cell and has at most two neighbors (its adjacent cells). The goal component of each node is unused; all necessary TM information is encoded in the response field.

We define the set of possible responses, $\mathcal{R}$, to be the set of all TM tape configurations plus two special symbols for the ends of the tape: $\mathcal{R} = (S_{tm} \times \Gamma_{tm} \times \{L, R\}) \cup \{\varepsilon_L, \varepsilon_R\}$. A standard response thus contains a state, a tape symbol, and a direction. The special responses, $\varepsilon_d = (\varepsilon, \varepsilon, d)$ for $d \in \{L, R\}$, are used to mark the dynamically created boundaries of the tape.

The initial TM tape is configured on a linear chain of Quest Graph nodes, with the focus node representing the TM's head. The setup involves three steps:

1. The initial tape symbols are written to the response field of each corresponding node.

2. Next, the direction symbols are initialized to point *inward* toward the focus: every node on the left has its direction set to $R$, and every node on the right has its direction set to $L$.

3. Finally, the TM's start state, $s_{tm}^0$, is placed on a single node adjacent to the focus (e.g., the left neighbor). To ensure the agent can identify this node, the focus node's own direction is set to be the opposite of that neighbor's direction. The state fields on all other nodes are arbitrary. For instance, a valid initial configuration is: $[(\cdot, \cdot, R), (s_{tm}^0, \cdot, R), (\cdot, \cdot, L), (\cdot, \cdot, L), (\cdot, \cdot, L)]$ where the middle node is the focus, and dots are wildcards.

During the simulation, the agent's function is executed at each step. It reads its local context (the focus node and its one or two neighbors) and determines the appropriate action based on two cases:

- First, if the agent is at an end of the tape (i.e., the focus node has a degree of 1), its task is to extend the tape. It performs a discover action, creating a new node with an empty goal and a special response $\varepsilon_d$, where $d$ is the direction opposite to the existing neighbor.

- Otherwise, if the focus node is in the middle of the tape, the agent simulates a standard TM transition as follows:

  1. Read State and Symbol: The agent reads the current tape symbol from the focus node's response. It then identifies the *previous* node by applying a simple rule: the previous node is the neighbor whose direction symbol is the opposite of the focus node's own direction symbol. The TM's current state is then read from that previous node's response.
  2. Compute and Write: With the current state and tape symbol, the agent emulates the TM's transition function, $\delta_{tm}$, to get the new state, new symbol, and new move direction. It then updates the focus node's response with these three new values.
  3. Move: Finally, the agent performs a move focus action, shifting the focus to the neighbor indicated by the new move direction.

This process repeats until the agent reads a state that is in the TM's set of accepting states, at which point the simulation halts. The agent's complete logic is formalized in Table 2.

To assert the correctness of the simulation, we show that these invariant patterns hold throughout:

- The current tape symbol can be identified from the focus node's response. This is ensured because the agent writes the new tape symbol to the focus node before moving.

- The current state of the TM is always stored in the response of one of the focus node's neighbors and can be unambiguously identified using the direction symbol on the focus node. This is because the configuration of direction symbols of the local context [left neighbor, focus node, right neighbor] always matches one of two patterns: $[R, L, L]$ or $[R, R, L]$. This predictability stems from the fact that the TM's head moves one cell at a time and the tape is not cyclic. If the focus node's response has the direction $L$, it means the last visit to this node ended with a left move; this, in turn, means the agent must have come from the right side, and vice versa. Hence, the previous node is always the neighbor whose direction symbol is opposite to that of the focus node.

- The degree of each node is at most two, ensuring the tape remains linear. This is guaranteed because the agent only creates a new node when it is at an end of the tape.

Given that $S_{tm}$ and $\Gamma_{tm}$ are finite, each Quest Graph node involved in the Turing machine simulation interacts with at most two neighbors, and the agent is stateless (deriving all necessary information from the Quest Graph via the focus node and its two neighbors as per Table 2), this construction faithfully simulates an arbitrary Turing machine. Therefore, the Quest Graph framework, with such an agent function, is Turing complete. □

While this proof uses a specialized Quest Graph where nodes have at most two neighbors, practical applications will likely involve graphs with higher degrees of connectivity. This richer connectivity does not reduce the framework's expressiveness, provided the agent can still simulate the base $C = 2$ case (for example, by focusing on a "tape-like" path within a more complex graph).

| $v_f$, Neighbor 1, Neighbor 2 | Return action |
|---|---|
| $(\cdot, t, R), (s^1, \cdot, R), (s^2, \cdot, L)$ | response $\delta_{\text{tm}} : (s^2, t) \mapsto (s', t', d')$ and move $d'$ |
| $(\cdot, t, L), (s^1, \cdot, R), (s^2, \cdot, L)$ | response $\delta_{\text{tm}} : (s^1, t) \mapsto (s', t', d')$ and move $d'$ |
| $(\cdot, t, R), (s^1, \cdot, L), (s^2, \cdot, R)$ | response $\delta_{\text{tm}} : (s^1, t) \mapsto (s', t', d')$ and move $d'$ |
| $(\cdot, t, L), (s^1, \cdot, L), (s^2, \cdot, R)$ | response $\delta_{\text{tm}} : (s^2, t) \mapsto (s', t', d')$ and move $d'$ |
| $(\cdot, t, R), (\cdot, \cdot, R), (f, \cdot, L)$ | stop |
| $(\cdot, t, L), (f, \cdot, R), (\cdot, \cdot, L)$ | stop |
| $(\cdot, \cdot, \cdot), (\cdot, \cdot, R), -$ | discover with response $\varepsilon_L$ and link to focus node |
| $(\cdot, \cdot, \cdot), (\cdot, \cdot, L), -$ | discover with response $\varepsilon_R$ and link to focus node |

Table 2: The agent function for simulating an arbitrary Turing machine. The dots $(\cdot)$ represent wildcards, with variables defined as follows: $s^1, s^2, s' \in S_{\text{tm}}$; $t, t' \in \Gamma_{\text{tm}}$; $d' \in \{L, R\}$; and $f \in F_{\text{tm}}$.

## B  COMPUTABILITY OF QUEST DECISION PROCESS

We formally define the Quest Decision Process (QDP) and its finite version.

**Definition 12.** *A **Quest Decision Process** (QDP) is a more constrained variant of the Quest Graph (Appendix A), formalized as a tuple analogous to a POMDP: $(\mathcal{V}_{qdp}, \mathcal{G}, \chi, R_{qdp}, \Omega_{qdp}, \mathcal{O}_{qdp}, \gamma)$. The key distinction from a standard POMDP is that the QDP operates deterministically. Instead of a probabilistic transition function, it uses a deterministic agent function, $\chi$, which maps a local context directly to an action. The components are defined as follows:*

- *$\mathcal{V}_{qdp}$: A finite set of states, corresponding to an augmented set of Quest Graph nodes.*

- *$\mathcal{G}$: A finite set of actions, corresponding to the set of goals in the Quest Graph.*

- *$\chi$: The deterministic agent function, which replaces the standard transition function.*

- *$R_{qdp}, \Omega_{qdp}, \mathcal{O}_{qdp}, \gamma$: The standard reward function, observation set, observation function, and discount factor, respectively.*

*Each quest node $v \in \mathcal{V}_{qdp}$ is a tuple $(g, r)$, where $g \in \mathcal{G}$ is a goal and the response $r \in \mathcal{R} \cup \{\pi\}$. The response set is augmented with a special **parent mark**, $\pi$. A node with this mark is a parent of the current focus node. For brevity, we will refer to the QDP state set as $\mathcal{V}$.*

*The QDP operates via four primary hierarchical actions:*

- *Discover input: Add a new input node as a child of the current focus node.*

- *Discover sub-quest: Add a new quest node as a child of the current focus node, mark the current focus node as a parent, and shift the focus to the new child.*

- *Complete quest: Use respond then move action to record the result, and move the focus back to its parent.*

- *Stop: Halt the computation. This action is only permitted when the focus node has no parent.*

*A **finite QDP** (FQDP) is a QDP that further restricts each node to a finite number of children. When this limit is reached, the agent can only perform actions that do not create additional children for the focus node.*

By restricting the agent in this manner, the FQDP is no longer Turing complete. Instead, it becomes computationally equivalent to a less powerful, yet still significant, model of computation:

**Theorem 13.** *A Finite Quest Decision Process (FQDP) is computationally equivalent to a deterministic pushdown automaton (DPDA).*

Let a pushdown automaton (PDA) be defined as a 7-tuple $(S_{\text{pda}}, \Sigma_{\text{pda}}, \Gamma_{\text{pda}}, \delta_{\text{pda}}, s^0_{\text{pda}}, Z^0_{\text{pda}}, F_{\text{pda}})$, where:

- *$S_{\text{pda}}$ is a finite set of states.*

- $\Sigma_{\text{pda}}$ is a finite set of input symbols (the input alphabet).

- $\Gamma_{\text{pda}}$ is a finite set of stack symbols (the stack alphabet).

- $\delta_{\text{pda}}$ is the transition function.

- $s_{\text{pda}}^0 \in S_{\text{pda}}$ is the initial state.

- $Z_{\text{pda}}^0 \in \Gamma_{\text{pda}}$ is the initial stack symbol.

- $F_{\text{pda}} \subseteq S_{\text{pda}}$ is the set of accepting states.

A PDA's configuration is a tuple $(s, w, t)$, where $s \in S_{\text{pda}}$ is the current state, $w \in \Sigma_{\text{pda}}^*$ is the remaining input, and $t \in \Gamma_{\text{pda}}^*$ is the current stack content. The transition function for a general (non-deterministic) PDA is defined as $\delta_{\text{pda}} : S_{\text{pda}} \times (\Sigma_{\text{pda}} \cup \{\varepsilon\}) \times \Gamma_{\text{pda}} \to \mathcal{P}_{\text{fin}}(S_{\text{pda}} \times \Gamma_{\text{pda}}^*)$, mapping the current state, input symbol (or $\varepsilon$), and top of the stack to a finite set of possible next states and stack operations.

There is an equivalent alternative definition where the function only allows a single symbol to be pushed or popped at a time:

$$\delta_{\text{pda}} : S_{\text{pda}} \times (\Sigma_{\text{pda}} \cup \{\varepsilon\}) \times \Gamma_{\text{pda}} \to \mathcal{P}_{\text{fin}}(S_{\text{pda}} \times (\Gamma_{\text{pda}} \cup \{\varepsilon\}))$$

For the proofs that follow, we will use this latter, more constrained definition.

A PDA accepts an input string if, after the entire string is processed, either of two conditions is met: 1. the machine is in an accepting state ($s \in F_{\text{pda}}$), or 2. the stack is empty.

A deterministic PDA (DPDA) is a PDA with two restrictions:

- For any given configuration, there is at most one possible transition.

- If an $\varepsilon$-transition is defined for a given state and stack symbol, no other transitions are defined for that same state and stack symbol.

We will prove this equivalence by constructing a bisimulation between the FQDP and a DPDA, showing the simulation in each direction. The FQDP's recursive, stack-like nature allows it to be modeled by a DPDA.

**Lemma 14.** *A deterministic pushdown automaton (DPDA) can simulate a Finite Quest Decision Process (FQDP).*

*Proof.* We construct a DPDA to simulate the FQDP by mapping the FQDP's components and actions to their DPDA equivalents.

We map FQDP components to DPDA components as follows:

- DPDA States: Each state of the DPDA represents a unique local context in the FQDP. We define a bijective mapping, $\sigma_{\text{pda}} : \mathcal{V}^{\leq C} \to S_{\text{pda}}$, from the finite set of all possible FQDP contexts to the DPDA's state set, $S_{\text{pda}}$. Since the FQDP's context is finite, its state set is also finite.

- DPDA Stack: The DPDA's stack mirrors the FQDP's call-return hierarchy. When the FQDP agent discovers a sub-quest, the state corresponding to the updated local context is pushed onto the stack.

The FQDP actions are simulated by the DPDA as follows:

- Discover input: The DPDA consumes an input symbol and transitions to a new state that reflects the FQDP's updated local context with the new input node.

- Discover sub-quest: The DPDA pushes the updated state, corresponding to the local context with a parent mark, onto the stack. It then performs an $\varepsilon$-transition to a new state representing the context of the newly created sub-quest.

- Complete quest: The DPDA pops the parent's state from the stack. It then performs an $\varepsilon$-transition to a new state that reflects the parent's updated context, which now incorporates the response from the completed sub-quest.

- Stop: The DPDA halts when it reaches a state that corresponds to the FQDP's termination condition.

A detailed mapping of these actions to formal transition rules is shown in Table 3.

The following invariants are maintained throughout the simulation:

- The states of the DPDA accurately represent the FQDP's local contexts.

- The symbol at the top of the DPDA's stack corresponds to the context of the parent node in the FQDP.

Because the FQDP is deterministic, the corresponding DPDA transitions can resolve unambiguously, ensuring these invariants hold.

This construction therefore demonstrates that a DPDA can faithfully simulate the behavior of an FQDP. $\qquad\square$

| FQDP Operation | Equivalent DPDA Transition |
| --- | --- |
| Discover input $i$ | $\delta_{\text{pda}} : (s, i, Z) \mapsto \left(\sigma_{\text{pda}}(\sigma_{\text{pda}}^{-1}(s) + i), Z\right)$ |
| Discover sub-quest $q$ | $\delta_{\text{pda}} : (s, \varepsilon, Z) \mapsto \left(\sigma_{\text{pda}}(q), \sigma_{\text{pda}}(\sigma_{\text{pda}}^{-1}(s) \leftarrow \pi)Z\right)$ |
| Complete quest with $r$ | $\delta_{\text{pda}} : (s, \varepsilon, pZ) \mapsto \left(\sigma_{\text{pda}}(\sigma_{\text{pda}}^{-1}(p) + \sigma_{\text{pda}}^{-1}(s) \leftarrow r), Z\right)$ |
| Stop at $v^{\leq C}$ | halt at $\sigma_{\text{pda}}(v^{\leq C}) = f$ |

Table 3: Mapping FQDP operations to DPDA transition rules. The notation is defined as follows: $\sigma_{\text{pda}}$ is an invertible mapping from an FQDP's local context to a DPDA state. $\sigma_{\text{pda}}^{-1}(\ldots)$ represents the inverse mapping from a DPDA state back to the corresponding FQDP context. $s, p \in S_{\text{pda}}$ are variables for the current and parent DPDA states, respectively. $f \in F_{\text{pda}}$ is a variable for an accepting DPDA state. $i, q \in \mathcal{V}$ are variables for FQDP input and sub-quest nodes, respectively. $r \in \mathcal{R}$ is a variable for the response at the focus node. $+$ denotes appending a new node to the local context. $v \leftarrow r$ denotes responding to a node $v$ with $r$. $v^{\leq C} \in \mathcal{V}^{\leq C}$ is a variable for an FQDP local context with at most $C$ nodes. $Z \in \Gamma_{\text{pda}}$ is a variable for the symbol at the top of the stack; $sZ$ denotes pushing state $s$ onto the stack.

**Lemma 15.** *A Finite Quest Decision Process (FQDP) with a local context size of $C = 4$ can simulate an arbitrary deterministic pushdown automaton (DPDA).*

*Proof.* We construct an FQDP to simulate an arbitrary DPDA. To achieve this, we encode the DPDA's configuration information directly into the structure of the FQDP's quest nodes.

An FQDP quest node is a tuple $(g, r)$. The goal, $g$, is itself a tuple, $(\text{type}, Z, s)$, representing the node's role in the simulation (type), a stack symbol from the DPDA ($Z \in \Gamma_{\text{pda}} \cup \{\varepsilon\}$), and a DPDA state ($s \in S_{\text{pda}} \cup \{\varepsilon\}$).

The response, $r \in S_{\text{pda}} \cup \Sigma_{\text{pda}} \cup \{\pi, \varepsilon, \varphi\}$, is used to pass operational results. These can be a DPDA state, an input symbol, the parent mark ($\pi$), an empty value ($\varepsilon$), or a special terminal response, $\varphi$, which indicates the DPDA has completed processing the input and the simulation is ready to terminate.

We define three node types to drive the simulation:

- $i$ (input node), for consuming an input symbol;

- $q$ (quest node), for initiating a sub-quest to simulate a stack push; and

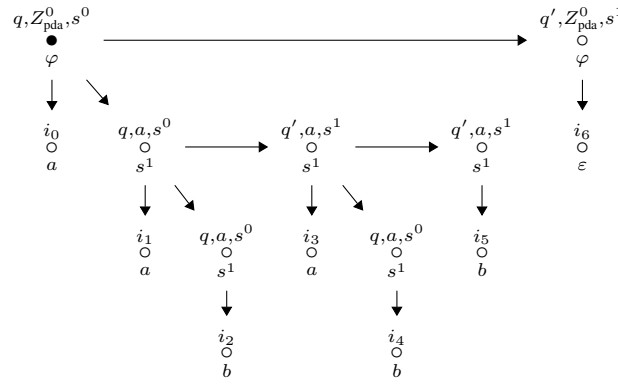

Figure 6: An FQDP with $C = 4$ simulating a DPDA that accepts the input string $aababb$. The sequence of input nodes (labeled $i_0, i_1, \ldots$) illustrates the order in which the DPDA consumes the input symbols. $\varphi$ is a special terminal response, indicating the DPDA has successfully processed the input. The arrows are for visualization only, showing the parent-child relationships created during the simulation, and do not represent the graph's undirected edges.

| last child type | perform |
| --- | --- |
| empty | Read Input: Discover a new "input node." If the DPDA can make an $\varepsilon$-transition, the new node's response may be empty. |
| input $a$ | Apply Transition: Compute $\delta_{\mathrm{pda}} : (s, a, Z) \mapsto (s', \gamma')$. If $\gamma'$ is a push operation, discover a "sub-quest" node representing $(s', \gamma')$, then respond with $\pi$ to mark the current node as a parent and move focus to the newly created sub-quest. Otherwise, if it is a pop operation, complete the current node by responding with the new state $s'$ and moving focus to the parent node. |
| sub-quest | Create Replica: Discover a new "replica node" using the state $s$ from the sub-quest's response and the stack symbol copied from the current focus node's goal. Then, respond with $\pi$ to mark the current node as a parent and move focus to the newly created replica node. |
| replica | Exit Replica: Respond with the state $s$ from the replica's response and move focus to the parent quest node. |

Table 4: Agent actions for simulating DPDA.

- $q'$ (replica node), a copy of a quest node created to ensure the agent's local context size does not exceed the limit of $C$.

The simulation begins with a quest node representing the DPDA's initial state and an empty stack. The agent's policy is then determined by the configuration of the focus node's children, as detailed in Table 4 and summarized here:

- No Children: The agent discovers a new input node to consume the next symbol from the DPDA's input tape.

- Last Child is an input node: The agent has the necessary information to compute the DPDA transition. If the transition is a "push", the agent discovers a new quest node and pursues it as a sub-quest. If it is a "pop", the agent completes the current node and moves the focus back to its parent.

- Last Child is a sub-quest node: This indicates a return from a "pop" operation. The agent discovers a replica node, using the state from the sub-quest's response and the stack symbol from the current focus node's goal, and then pursues the replica as a sub-quest.

- Last Child is a replica node: This indicates the agent is in the process of popping the stack. It continues to move up the parent chain until a non-replica quest node is reached.

This process continues until the DPDA's acceptance conditions are met, at which point the FQDP begins its termination sequence. The agent responds to the current quest with the special terminal symbol, $\varphi$, and moves focus to its parent. It repeats this process of moving up the parent chain until the root node is reached. Once the root node is marked with the terminal symbol, the agent performs a "stop" action, halting the simulation.

The invariant of this simulation can be understood from the perspective of the DPDA's transitions. The invariant state is defined as the point where the agent's focus is on a quest or replica node that has an input node as its most recently created child.

In this configuration, the following properties hold:

- The agent has all the necessary information to compute the next DPDA transition.

- The DPDA's current state and the symbol at the top of its stack are stored in the goal of the focus node.

- The next input symbol is stored in the response of the child input node.

- The maximum degree of the focus node never exceeds four, ensuring the local context size remains within the limit of $C = 4$.

From this state, the agent's policy is deterministic. It computes the transition and executes a sequence of actions that will eventually lead it to a new state that also satisfies this invariant (or to a halting condition).

Because the agent's function depends only on its local context (bounded by $C = 4$), this construction successfully simulates an arbitrary DPDA. $\qquad\square$

Based on the preceding two lemmas, which establish a simulation equivalence in both directions, we conclude that the FQDP is computationally equivalent to a DPDA.

## C    COMPUTABILITY OF NON-DETERMINISTIC FINITE QUEST DECISION PROCESSES

**Definition 16.** *A **Non-deterministic Finite Quest Decision Process** (NFQDP) is a variant of the FQDP, defined by five primary hierarchical actions:*

- *Discover input: Adds a new input node as a child of the current focus node. Input nodes are considered complete upon discovery.*

- *Discover sub-quest: Adds a new, incomplete quest node as a child of the current focus node.*

- *Pursue sub-quest: Marks the current focus node as a parent and shifts the focus to one of its incomplete child nodes.*

- *Complete quest: Marks the current focus node as complete by recording a result in its response, then moves the focus back to its parent. This action is only permitted after all of the focus node's children are complete.*

- *Stop: Halts the computation. This action is only permitted when the root node of the quest tree is complete.*

*The key distinction in the NFQDP is the separation of the discover and pursue sub-quest actions, which allows an agent to traverse a pre-existing graph structure depth-wise.*

The following corollary states the computational power of this new model.

**Corollary 17.** *A Non-deterministic Finite Quest Decision Process (NFQDP) is computationally equivalent to a pushdown automaton (PDA).*

First, we show that a PDA can simulate an NFQDP, as stated in the following lemma.

**Lemma 18.** *A pushdown automaton (PDA) can simulate a Non-deterministic Finite Quest Decision Process (NFQDP).*

*Proof.* We extend the proof in Appendix B by addressing the separation of the discover and pursue sub-quest actions in the NFQDP. The simulation of most actions remains the same, with the key changes outlined below:

- Discover sub-quest: The PDA performs an $\varepsilon$-transition to a new state that reflects the NFQDP's updated local context with the new quest node.

- Pursue sub-quest: The PDA pushes the current state (representing the parent's context with a parent mark) onto the stack. It then performs an $\varepsilon$-transition to a new state representing the context of the child sub-quest. This step leverages the non-determinism of the PDA to select the correct transition to match the existing local context of the pre-built tree structure.

- The other actions (Discover input, Complete quest, Stop) are simulated similarly to their counterparts in the FQDP-DPDA simulation.

A PDA's transition function can simulate an NFQDP by leveraging its inherent non-determinism. This is necessary to resolve the agent's state when it pursues a sub-quest, as the local context of the target node can vary depending on the pre-built graph structure. Because the set of possible local contexts in an NFQDP is finite by definition, a finite set of PDA transition rules can be constructed to cover every possible configuration. $\square$

For the other direction of the proof, a direct simulation of a PDA by an NFQDP is complicated by the PDA's non-deterministic $\varepsilon$-transitions. Therefore, we will instead establish the equivalence by showing that an NFQDP can be constructed to accept any string from any given context-free language (CFL), which is the class of languages recognized by a PDA (Hopcroft et al. (2001)).

A ***context-free grammar*** (CFG) is a formal grammar defined by the 4-tuple $(N_{\text{cfg}}, \Sigma_{\text{cfg}}, P_{\text{cfg}}, S_{\text{cfg}})$, where:

- $N_{\text{cfg}}$ is a finite set of non-terminal symbols.
- $\Sigma_{\text{cfg}}$ is a finite set of terminal symbols.
- $S_{\text{cfg}} \in N_{\text{cfg}}$ is the start symbol.
- $P_{\text{cfg}}$ is a finite set of production rules of the form $A \to \gamma$, where $A \in N_{\text{cfg}}$ is a non-terminal and $\gamma \in (N_{\text{cfg}} \cup \Sigma_{\text{cfg}})^*$ is a string of terminals and non-terminals.

A context-free language (CFL) is a language generated by a context-free grammar.

Every context-free grammar can be converted into an equivalent grammar in ***Chomsky normal form*** (CNF) (Chomsky (1959)), where all production rules must be in one of the following forms:

- $A \to BC$, where $A, B, C$ are non-terminal symbols.
- $A \to a$, where $A$ is a non-terminal and $a$ is a terminal symbol.
- $S_{\text{cfg}} \to \varepsilon$, where $S_{\text{cfg}}$ is the start symbol. This rule is only permitted if the language contains the empty string, and if this rule exists, the start symbol cannot appear on the right-hand side of any other rule.

**Lemma 19.** *A Non-deterministic Finite Quest Decision Process (NFQDP) with a local context size of $C = 3$ can be constructed to accept any given context-free language (CFL).*

*Proof.* Given an input string and a CFG in Chomsky Normal Form, we first pre-build a graph that represents the space of all possible parse trees for that string. This is feasible with a local context size

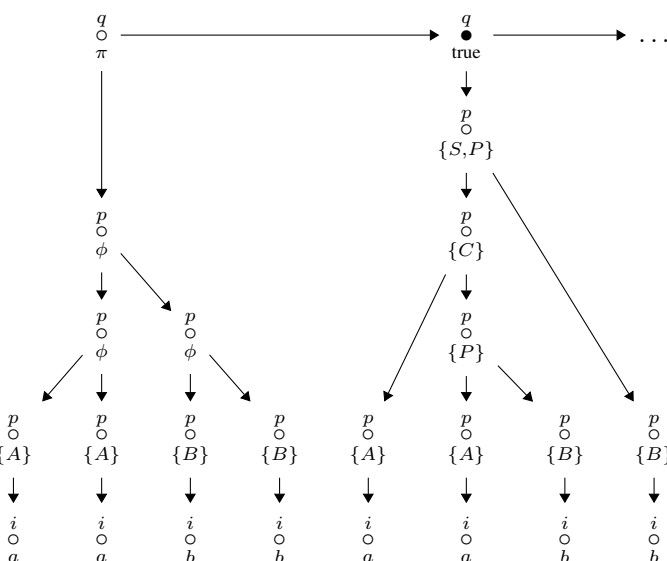

Figure 7: The parse graph for an NFQDP simulation on the input string $aabb$. The graph represents the space of all possible parse trees according to the following CFG in Chomsky Normal Form: $S \to CB \mid AB, P \to CB \mid AB, C \to AP, A \to a$, and $B \to b$.

of $C = 3$ because a CNF parse tree is a binary tree, meaning any node has at most two children and one parent. The agent's task is then to perform a depth-first traversal of this pre-built graph to find a parse tree that is valid according to the grammar's production rules.

The graph is structured as a chain of binary trees linked at their roots, where each binary tree represents a possible parse tree for the input string. We use three node types to construct this parse graph:

- $i$ (input node): An input node is a leaf in a parse tree. Its response contains the corresponding terminal symbol from the input string.

- $q$ (quest node): A quest node is the root of a parse tree. Its response is a boolean value indicating whether that parse or any of the subsequent parses is valid.

- $p$ (production node): A production node represents the application of a grammar rule. Its response contains the set of all non-terminal symbols that can generate the substring covered by its subtree.

The goal field for all nodes simply specifies the node's type from the set $\{i, q, p\}$. Figure 7 illustrates this construction for the input string $aabb$.

From the root node, the agent traverses the graph depth-wise, pursuing sub-quests to explore the different branches of the parse trees. At each production node, the agent checks the grammar for rules whose right-hand side is formed by the non-terminal symbols from the node's children. It then collects the left-hand side non-terminals of all such matching rules and stores them in its response field.

Upon returning to the root node of a parse tree, the agent pursues the next quest node in the chain to evaluate the next possible parse tree. This forward traversal continues until the agent has explored all possible parse trees in the graph.

At that point, a termination sequence begins, moving backward along the chain of quest nodes. For each quest node, the agent records a boolean in its response field. The response is set to "true" if its own parse tree was valid (i.e., its root contained the grammar's start symbol) or if the subsequent

quest node already has a response of "true". Otherwise, the response is "false". After this backward pass is complete, the agent reaches to the initial root of the entire graph and halts.

Because the grammar is in Chomsky Normal Form, any parse tree for the input string is a binary tree. This structure ensures that the total number of possible parse trees is bounded and is equal to the Catalan number corresponding to the input string's length. Furthermore, since the set of production rules is finite, the set of all non-terminal symbols that can appear in the response of any production node is also finite. These two conditions together guarantee that the pre-built graph has a bounded size and that the traversal process is guaranteed to terminate for any non-empty input string.

Finally, the special case of the empty string is handled directly. If the language contains the empty string, the agent can be designed to immediately halt and accept when the input is pre-built with a single quest node. This construction shows that an NFQDP can be built to accept any given CFL. ☐

## D  COMPUTABILITY OF REFERENCE-AUGMENTED FINITE QUEST DECISION PROCESSES

**Definition 20.** *A **Reference-Augmented Finite Quest Decision Process (RQDP)** is an FQDP augmented with three components that enable memory retrieval:*

- *A non-finite reference set, $\mathcal{T}$, whose elements serve as tags for nodes. Multiple nodes can share the same reference. This set is equipped with an equality operator, $\mathcal{T} \times \mathcal{T} \to \{true, false\}$.*

- *A **reference-generating function**, $\tau : \mathcal{T} \times \mathcal{G} \to \mathcal{T}$. When any new node is created, its reference is computed by applying $\tau$ to the parent node's reference and the new node's goal.*

- *A **retrieve** action, $\mathcal{V}^{\leq C} \to \mathcal{G}$. When the agent performs this action, it outputs a goal. The Quest Graph then creates a new node, attaches it to the focus node, and assigns it a reference using the $\tau$ function. It then uses this new reference to retrieve the most recently updated response associated with a matching reference in its history; this response is then assigned to the new node. If no such match exists, the new node's response is set to the empty symbol, $\varepsilon$.*

*A **Non-deterministic Reference-Augmented Finite Quest Decision Process** (NRQDP) is a variant of the RQDP that incorporates the non-deterministic, hierarchical actions of the NFQDP.*

We will now show that the RQDP is Turing complete. This is established by demonstrating that it can simulate a standard Turing machine with a one-way infinite tape (Hopcroft et al. (2001)). Because the non-deterministic NRQDP can simulate its deterministic counterpart, it follows that the NRQDP is also Turing complete.

**Theorem 21.** *An agent with a local context capacity of $C = 4$ operating under a Reference-Augmented Finite Quest Decision Process (RQDP) can simulate an arbitrary Turing machine.*

*Proof.* To simulate a TM, we first define the structure of the RQDP nodes used in the construction. Each node's goal is a tuple $(type, s, t)$, representing the node's type, a TM state $s \in S_{tm}$, and a tape symbol $t \in \Gamma_{tm}$. The response is a tuple $(s', t')$, representing the next TM state and the symbol to be written. We define four node types to drive the simulation:

- $q$ (quest node): For initiating a sub-quest, which simulates moving the tape head right.

- $q'$ (replica node): A copy of a quest node, used to manage the local context size, similar to its role in the FQDP-DPDA simulation.

- $f$ (fetch node): For retrieving a tape symbol.

- $w$ (write node): For recording a tape symbol.

The simulation strategy uses this machinery to represent the TM's tape in two distinct ways: the portion of the tape to the left of the head is maintained by the FQDP's stack-like hierarchy, while the portion to the right is managed by the reference and retrieval system.

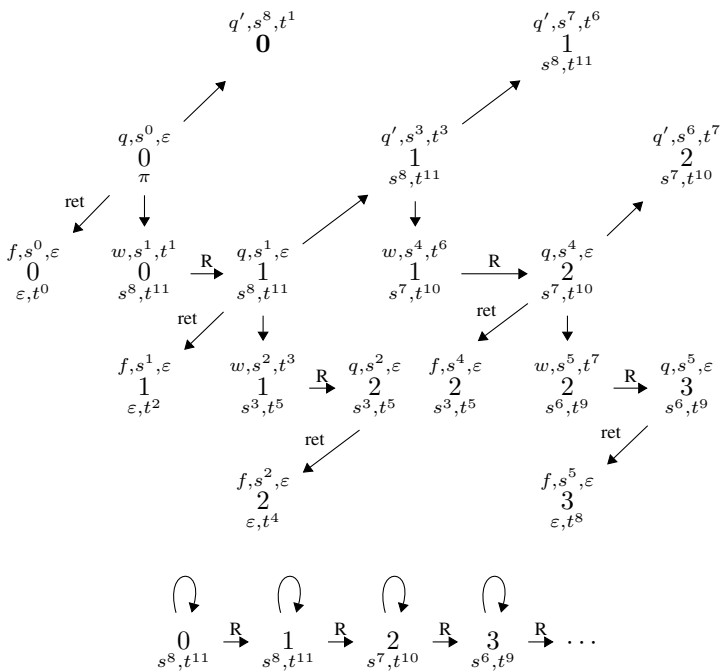

Figure 8: An RQDP with $C = 4$ simulating a TM. Top: The Quest Graph execution trace, where parent-child arrows are for visualization only and do not represent the graph's undirected edges. Edges labeled "ret" indicate a retrieve action, while "R" corresponds to a rightward move of the TM head. The depicted simulation follows the move sequence $RRLRRLLL$, starting from the quest node marked with a parent symbol ($\pi$) and ending at the node with the bold reference. Bottom: The corresponding reference graph, where each numbered node stores the cached response for that reference. In the diagrams, the state sequence is denoted by $s^0, s^1, \ldots$ and tape symbols by $t^0, t^1, \ldots$, in order of their creation. The diagonal layout is a stylistic choice to intuitively align rightward movement with the rightward direction on the page and to ensure retrieve action arrows point clearly to the most recently updated nodes.

We define the reference set to be the set of tape positions, $\mathbb{Z}^+$, where a reference $\xi \in \mathbb{Z}^+$ denotes the current tape position. The reference-generating function, $\tau$, is defined to simulate the rightward movement of the tape head. When the agent creates a new quest node (type $q$), the function increments the reference to the next tape position: $\tau : (\xi, (q, \cdot, \cdot)) \mapsto \xi + 1$. The $\cdot$ symbol is a wildcard, indicating this rule applies regardless of the state or tape symbol in the goal. For all other node types or conditions, the reference remains unchanged.

The agent's policy is a set of deterministic rules based on the type of the focus node and the state of its children. The simulation cycle can be constructed as follows:

1. When the focus is on a quest node: The agent's primary goal is to gather the information needed to compute a TM transition.

   (a) If the node has no children, the agent first needs to read the current tape symbol. It performs a retrieve action to create a fetch node. According to the reference-generating function, this new fetch node is assigned the same reference as its parent quest node. This ensures that the system retrieves the symbol from the most recently updated response of a node with that reference, correctly fetching the tape symbol at the current head position.

   (b) If the node has a fetch node as a child, the agent now has both the state $s$ (from its own goal) and the tape symbol $t$ (from the fetch node's response). It computes the

TM transition $\delta_{\text{tm}} : (s, t) \mapsto (s', t', d')$. The subsequent action depends on the move direction, $d' \in \{L, R\}$:

- If $d' = R$ (move right), the agent discovers a write node to store the new symbol, with the goal set to $(w, s', t')$, it then moves the focus to this new write node.
- If $d' = L$ (move left), the agent responds to the current node with $(s', t')$, which completes the node. It then moves the focus to its parent.

(c) If the node has a write node as a child, this indicates that the agent is executing a leftward move. It discovers a new replica node with $(q', s, t)$ as the goal, where $s$ is from the response of the write node, and $t$ is from the goal of the write node. Then, it pursues this replica node as a sub-quest.

(d) If the node has both a write node and a replica node as children, this indicates that the agent is returning from a replica node (a leftward move). The agent responds with the result from the completed replica node and moves the focus to its own parent, continuing the leftward sequence.

2. When the focus is on a replica node, most cases are similar to the quest node:

(a) If the node has no children, the agent uses the state and tape symbol from its goal to compute the TM transition and then performs the same actions described in item 1b.

(b) The rest of the cases are the same as those described in items 1c and 1d.

3. When the focus is on a write node: The agent's goal is to initiate the next rightward step of the simulation.

(a) If the node has no children, it discovers a quest node with the goal $(q, s, \varepsilon)$, where $s$ is from the write node's goal. It then moves the focus to this new quest node to begin the next simulation cycle.

(b) If it already has a quest node child, this signifies a return from a deeper part of the computation (a leftward move). The agent responds with the result from the completed child and moves the focus to its own parent, continuing the leftward sequence.

This cycle continues until the TM reaches a halting state, at which point the RQDP uses the same termination procedure as the FQDP to halt the computation. Figure 8 illustrates an RQDP with $C = 4$ simulating a TM.

The initial tape configuration is established in a setup phase. The process begins with a single quest node at reference $\xi = 0$. From this starting point, a special initializer agent populates the graph with the input string. It does so by repeatedly executing the rightward-move sequence from the simulation procedure described above, with one modification: the retrieve action is replaced with a discover input action to write the tape symbols directly onto the newly created nodes. Once this initialization is complete, the main simulation agent takes over, with its focus starting at the final quest node in the chain, which corresponds to the rightmost end of the initial tape.

The invariant of this simulation is defined from the perspective of the TM's transitions. The invariant state is the point at which the agent's focus is on either a replica node, or a quest node that has a fetch node as its most recently created child.

In this configuration, the following properties hold:

- The agent has all the necessary information to compute the next TM transition.

- The TM's current state is stored in the goal of the focus node.

- The tape symbol at the current head position is retrieved based on the focus node's type: if the focus is a quest node, the symbol is in the response of its fetch child; if the focus is a replica node, the symbol is in the goal of the replica node itself.

- The maximum degree of the focus node never exceeds four, ensuring the local context size remains within the limit of $C = 4$.

From this state, the agent's policy is deterministic. It computes the transition and executes a sequence of actions that eventually leads to a new state that also satisfies this invariant, or to a halting condition.

This construction shows that an RQDP with $C = 4$ can simulate a TM. □

## E  COMPUTABILITY OF LANGUAGE MODEL

Let a language model be defined as a tuple $LM = (\Gamma_{\text{lm}}, \delta_{\text{lm}}, C)$, where:

- $\Gamma_{\text{lm}}$ is a finite set of tokens (the vocabulary).
- $\delta_{\text{lm}} : \Gamma_{\text{lm}}^C \to \Gamma_{\text{lm}}$ is the transition function, which maps a sequence of $C$ tokens to the next token.
- $C \in \mathbb{Z}^+$ is the context length, which is finite.

A finite state machine (FSM) can be defined as a tuple $(S_{\text{fsm}}, \Sigma_{\text{fsm}}, \delta_{\text{fsm}}, s_{\text{fsm}}^0, F_{\text{fsm}})$, where:

- $S_{\text{fsm}}$ is a finite set of states.
- $\Sigma_{\text{fsm}}$ is a finite set of input symbols (the input alphabet).
- $\delta_{\text{fsm}} : S_{\text{fsm}} \times \Sigma_{\text{fsm}} \to S_{\text{fsm}}$ is the transition function.
- $s_{\text{fsm}}^0 \in S_{\text{fsm}}$ is the initial state.
- $F_{\text{fsm}} \subseteq S_{\text{fsm}}$ is the set of accepting states.

**Lemma 22.** *A language model (LM) with a context size of $C = 2$ can simulate an FSM.*

In practice, an LM's context is updated by appending user inputs. For the purpose of this simulation, we adopt this same mechanism to feed input symbols from the FSM into the LM's context.

*Proof.* Given an FSM, we can construct an LM with a context size of $C = 2$ as follows. First, we define the LM's vocabulary as the union of the FSM's states and input alphabet: $\Gamma_{\text{lm}} = S_{\text{fsm}} \cup \Sigma_{\text{fsm}}$. Second, we define the LM's transition function $\delta_{\text{lm}}$ to perfectly emulate the FSM's transition function, such that for any state $s \in S_{\text{fsm}}$ and input symbol $a \in \Sigma_{\text{fsm}}$, we set $\delta_{\text{lm}}(s, a) = \delta_{\text{fsm}}(s, a)$.

The simulation begins with the FSM's initial state, $s_{\text{fsm}}^0$, in the context. For each symbol of the input string, the process alternates between two steps:

- First, the current input symbol is appended to the context.

- Second, the LM applies its transition function to the context (which now contains the current state and input symbol) to produce and append the next state.

An input string is considered accepted if the final state in the context after processing the entire string is in $F_{\text{fsm}}$.

This construction shows that an LM with $C = 2$ can simulate an FSM. □

**Lemma 23.** *There exists a finite state machine (FSM) that can simulate a language model (LM).*

*Proof.* Let $\sigma_{\text{lm}} : \Gamma_{\text{lm}}^C \to S_{\text{fsm}}$ be a bijective mapping from the set of all possible LM contexts to the FSM's state set, $S_{\text{fsm}}$. Since both the LM's vocabulary ($\Gamma_{\text{lm}}$) and context length ($C$) are finite, the state set $S_{\text{fsm}}$ is also finite. For this construction, we set the FSM's input alphabet to be the LM's vocabulary, $\Sigma_{\text{fsm}} = \Gamma_{\text{lm}}$.

We define the FSM's transition function, $\delta_{\text{fsm}}$, to simulate the LM's two primary actions:

- Autoregressive Generation: When the LM generates a token internally from its current context $(t_1, \ldots, t_C)$ to produce $t_{C+1}$, the FSM performs an epsilon-transition from state $\sigma_{\text{lm}}(t_1, \ldots, t_C)$ to the new state $\sigma_{\text{lm}}(t_2, \ldots, t_C, t_{C+1})$.

- Consuming External Input: When an external input token $a$ is appended to the LM's context, the FSM transitions on that symbol. From state $\sigma_{\text{lm}}(t_1, \ldots, t_C)$, the FSM reads input $a$ and transitions to the new state $\sigma_{\text{lm}}(t_2, \ldots, t_C, a)$.

The FSM's initial state is defined by the LM's initial context. Through these transitions, the FSM's state at any point perfectly tracks the LM's context. This construction shows that an FSM can simulate the behavior of a finite-context LM. □

With the previous two lemmas, we can conclude that

**Theorem 24.** *A language model with finite context is computationally equivalent to a finite state machine (FSM).*

## F  ANALYSIS OF BOUNDED MAXIMUM-DEPENDENCY COMPUTATION GRAPHS

A **Maximum-Dependency Computation Graph** (MCG) is a directed acyclic graph whose nodes are uniquely ordered, such that every node receives an incoming edge from all nodes with a lower order.

**Lemma 25.** *For any computation graph, there exists an equivalent Maximum-Dependency Computation Graph (MCG) that preserves all original dependencies and can be constructed by adding at most one node.*

*Proof.* The construction involves creating a total ordering of the nodes and then adding edges to form the MCG.

1. First, we establish an ordering by identifying all terminal nodes (those with no outgoing edges). If more than one exists, a new, single terminal node is created, connected to all original ones, and assigned order $0$.

2. We then traverse the graph backward: direct predecessors of order-$k$ nodes are assigned order $k + 1$, repeating until all nodes are ordered. If a node is already ordered, choose the maximum of the existing order and the new order. This process is guaranteed to terminate as the graph is a DAG. Any nodes sharing the same order are then assigned an arbitrary sub-order to create a total ordering.

3. Finally, for every pair of nodes, if one precedes the other in the total ordering and an edge does not already exist, one is added.

This process creates an MCG where every node depends on all preceding nodes. Crucially, it preserves the original graph's precedence, as our backward-traversal ordering ensures that if an edge existed originally, the source node is guaranteed to precede the target node in the final ordering. □

A **Bounded Maximum-Dependency Computation Graph** (BMCG) is an augmented version of an MCG with the following properties:

- There is a constant, $C \in \mathbb{Z}^+$, that defines the maximum allowed in-degree (number of dependencies) for any node.
- Any node in the original MCG with an in-degree greater than $C$ is transformed into a hierarchical tree of proxy nodes. This structure connects the original node to all of its dependencies while ensuring that the in-degree of any node in the resulting graph is no greater than $C$.

The total number of added proxy nodes and edges in a BMCG is a function of two parameters: the total number of nodes ($N$) in the original MCG and the maximum in-degree constraint ($C$). The detailed analysis of this relationship is as follows:

Let $k(d)$ denote the number of proxy nodes required for a node with in-degree $d$ (see Table 5).

$$k(d) = \begin{cases} \left\lfloor \frac{d-2}{C-1} \right\rfloor, & \text{if } d \geq 2 \\ 0, & \text{otherwise} \end{cases}$$

Let $K = k(N-1)$ denote the number of proxy nodes for the node with the maximum possible in-degree of $N-1$.

$$K = \left\lfloor \frac{N-3}{C-1} \right\rfloor$$
$$= O(N)$$

Let $N^+$ be the total number of added proxy nodes in the BMCG.

$$N^+ = \sum_{d=0}^{N-1} k(d)$$
$$= (C-1)\left(\frac{K(K+1)}{2}\right) - K\left[(C-1)K - [(N-1)-C]\right]$$
$$= K(N-2) - (C-1)\left(\frac{K(K+1)}{2}\right)$$
$$= O(N^2)$$

| $C$ | $d=1$ | 2 | 3 | 4 | 5 | 6 | 7 | 8 | 9 | 10 | 11 | $\ldots$ |
|---|---|---|---|---|---|---|---|---|---|---|---|---|
| 2 | 0 | 0 | 1 | 2 | 3 | 4 | 5 | 6 | 7 | 8 | 9 | |
| 3 | 0 | 0 | 0 | 1 | 1 | 2 | 2 | 3 | 3 | 4 | 4 | $\left\lfloor \frac{d-2}{C-1} \right\rfloor$ |
| 4 | 0 | 0 | 0 | 0 | 1 | 1 | 1 | 2 | 2 | 2 | 3 | |

Table 5: The number of proxy nodes required for a single $d$-degree node, shown for various context size constraints ($C$) and original node in-degrees ($d$).

Therefore, the final size of the resulting BMCG can be determined. Since the original MCG contains $N$ nodes and $O(N^2)$ edges, and the number of added proxy nodes ($N^+$) is also $O(N^2)$, the final graph is composed of $N + N^+ = O(N^2)$ nodes and $\frac{N(N-1)}{2} + N^+ = O(N^2)$ edges.

