# OpenReview forum: "Computability of Agentic Systems"
_ICLR.cc/2026/Conference — Submitted to ICLR 2026_

### Official Review · Reviewer_6Fk5 · 2025-10-31

**Soundness:** 3
**Presentation:** 3
**Contribution:** 3
**Rating:** 6
**Confidence:** 3

**Summary:**

This paper introduces a formal framework called the Quest Graph to analyze the computational power of deterministic agentic systems that are constrained by a finite context window. The central idea is to decouple the agent's reasoning module (modeled as a stateless function) from its working memory (modeled as a dynamic graph).

The authors establish a baseline that a standard Language Model (LM) with a finite context is computationally equivalent to a Finite State Machine (FSM). They then introduce and analyze variants of their Quest Graph framework:

* General Quest Graph: An unrestricted model where the agent can freely modify the graph. This is shown to be Turing complete.

* Quest Decision Process (QDP): A constrained, forward-only version designed to model hierarchical reasoning (like chain-of-thought). The Finite QDP (FQDP), where nodes have a finite number of children, is shown to be equivalent to a Pushdown Automaton (PDA).

* Reference-Augmented QDP (RQDP): This model enhances the FQDP with a retrieval mechanism (inspired by RAG) that allows the agent to access its own unbounded, out-of-context history.

The paper's main results are a series of theorems and corollaries that build a computational hierarchy for agentic systems.

* Result 1 (Baseline): A standard Language Model with a finite context window is computationally equivalent to a Finite State Machine (FSM) (Theorem 1).

* Result 2 (General Framework): The general, unrestricted Quest Graph framework, where an agent can modify nodes, is Turing complete (Theorem 2).

* Result 3 (Hierarchical Reasoning): A FQDP, which models deterministic, hierarchical, forward-only reasoning, is computationally equivalent to a DPDA (Theorem 3). A Non-deterministic FQDP (NFQDP) is equivalent to a general PDA (Corollary 17).

* Result 4 (Memory/Retrieval): The RQDP, which adds a mechanism to retrieve from an unbounded history, is Turing complete (Theorem 4). This demonstrates that a retrieval-like mechanism is sufficient to move from context-free computation back to universal computation.

* Result 5 (Efficiency Hierarchy): When simulating a general computation graph of size $N$ (which is transformed into a Bounded MCG of size $O(N^2)$):

  * LMs (FSMs) cannot simulate the graph (Corollary 9).
  * FQDP/NFQDPs (PDAs) can, but in exponential time ($O(2^N)$), as they must re-compute shared dependencies (Corollary 8).
  * RQDPs/Quest Graphs (TMs) can in polynomial time ($O(N^2 \log N)$ or $O(N^2)$), as they can retrieve or access previously computed results (Corollaries 6 & 7).

**Strengths:**

* The Quest Graph is a clean and powerful abstraction. Decoupling the stateless agent function (the LM core) from the stateful graph memory (the scratchpad/context) is an insightful way to model these systems and aligns well with modern architectures.

* Practical Relevance: The paper wisely does not stop at pure computability. The analysis in Section 6 is a key contribution. It translates the abstract theoretical hierarchy into concrete performance trade-offs (exponential vs. polynomial time), providing a compelling argument for why retrieval-augmented (RQDP-like) agents are more powerful than simple hierarchical (FQDP-like) ones for tasks with complex dependencies.

**Weaknesses:**

* The entire framework models the agent as a deterministic function (equivalent to an LM with temperature 0). This is a necessary simplification for a classic computability analysis, but it sidesteps the fundamentally stochastic nature of real-world LMs. The paper's conclusions apply only to this deterministic, best-case reasoning.

* The analysis assumes the agent function $\chi$ (the LM forward pass) takes constant time ($O(1)$). In practice, this is the most computationally expensive part of the entire system. This doesn't invalidate the analysis of the number of steps, but it's an important abstraction that could be misleading if applied too literally to wall-clock time.

**Questions:**

* The entire framework rests on the agent being deterministic. How does this hierarchy change when considering stochastic agents, which is the standard for real-world LMs? Does stochasticity simply map to non-determinism (like the NFQDP), or does it introduce a fundamentally different computational class?

---

> ### Author Response · Authors · 2025-11-16
>
> We thank the reviewer for these insightful questions and appreciate the opportunity to clarify our model's abstractions.
> The reviewer is correct to point out the simplification in our O(1) time assumption for the agent function, as this is the main computational bottleneck in practice.
> Our intention in Section 6 was not to model total wall-clock time, but to analyze the theoretical efficiency in terms of the number of steps or nodes visited required by each computational class (e.g., exponential vs. polynomial).
> We believe this distinction reveals fundamental scaling properties.
> To avoid confusion with real-world runtimes, we have revised the manuscript to use the term "theoretical efficiency" throughout.
> (This point is also discussed in our response to Reviewer 3.)
>
> Regarding the stochastic variant, we agree that a full analysis would require substantial work beyond this paper's scope.
> We distinguish between training, where stochasticity is beneficial for exploration,
> and our paper's focus, which is the computational power of the deterministic inference-time policy.
> We would like to justify this deterministic focus with two points:
>
> 1.  Low-temperature decoding (near-determinism) is empirically associated with better reasoning performance, as shown in “Exploring the Impact of Temperature on Large Language Models: Hot or Cold?” (Li et al., 2025).
>
> 2.  In practice, small non-zero temperatures are often used to ensure reproducibility and prevent autoregressive repetition (e.g., “the the the…”), rather than to induce meaningful stochastic branching.
> This addresses degeneracy rather than altering the underlying reasoning path. For these reasons, a deterministic abstraction captures the essential structure of how today’s agentic LLMs perform multi-step reasoning.
>
> We agree that understanding when stochasticity meaningfully expands an agent’s computational class (e.g., to BPP, PP, RP, ZPP) is an interesting and important open problem.
> We view this as a promising direction for future work and are grateful to the reviewer for highlighting it.
>
> --------
>
> This response was drafted by the author and edited paragraph-by-paragraph by an AI for readability only.

---

### Official Review · Reviewer_Hqpp · 2025-11-01

**Soundness:** 2
**Presentation:** 3
**Contribution:** 2
**Rating:** 4
**Confidence:** 4

**Summary:**

The paper develops a formal framework, the Quest Graph, to analyze the computability of deterministic agentic systems with finite context. It introduces constrained variants, QDP/FQDP/NFQDP for hierarchical reasoning and RQDP/NRQDP for reference-augmented retrieval, and maps them to classical automata classes: finite-context LMs ≅ FSMs, FQDP ≅ DPDA (context-free), and RQDP regains Turing completeness via a non-finite reference space and a retrieve action. They also analyze these architectures from a practical lens.

**Strengths:**

The theoretical core is the paper’s main asset. The formalization is careful, with clear separation between agent function and external working memory, and a neat alignment to the formal language hierarchy that makes the progression from LM→QDP→RQDP easy to follow. The hierarchy yields crisp claims: finite-context LMs are only regular; enforcing hierarchical “forward only” structure collapses power to pushdown; adding a reference mechanism with a non-finite tag space restores Turing completeness. The computation-graph analysis is a welcome bridge from theory to practice, turning abstract differences into complexity statements that are simple to reason about in real coordination workloads.

**Weaknesses:**

First and foremost, There is no empirical validation or case study demonstrating that the complexity separations manifest on realistic agent stacks; without at least small controlled experiments, it is hard to judge how these elegant results transfer outside the formal model. Even without the coverage of experiments, their is a clear lack of discussion around the existing architectures and the theory introduced. The overall presentation undersells where present-day agent systems actually fail and what actionable design guidance follows for the existing LLM based architectures. Given that the paper's main contribution is around theory, I would expec to have some more discussion around the practicality of their approaches.

Apart from its theoretical contributions, the overall writing of the paper is not very strong. The following suggestions focus not on minor issues such as grammar or phrasing, but on broader improvements that could enhance the overall clarity and presentation of the work.:
* Table 1, which summarizes the core architectures and their computational relationships, should appear much earlier, preferably by the end of Section 2.
* The paper would benefit from explicitly tying the proposed theoretical constructs (e.g., LMs, QDP, RQDP) to existing model classes or real LLM-based agent systems to make the abstractions more relatable.
* Several figures (e.g., Figures 1 and 2) convey overlapping information and could be merged into a single, more comprehensive diagram illustrating both the architectural flow and hierarchy.

Overall this is a theoretically insightful and timely paper that gives the community a shared language for the computability of agentic systems. But the paper lacks in overall clarity in writing and providing actionable insights/experimental evidence to support their theoretical claims, which leads me to lean towards rejection.

**Questions:**

Q1. How do existing AI systems map to the different architectures introduced in the paper? Which of these architectures are already implicitly represented in current models, and which ones remain purely theoretical and hence pose as a direction for future works

Q2. How do the theoretical limitations identified for each architecture (e.g., context bounds, hierarchical reasoning constraints, or retrieval assumptions) translate into practical bottlenecks observed in today’s agentic or LLM-based systems?

Q3. The paper focuses primarily on theoretical constructs and asymptotic analysis, could small-scale empirical experiments or simulations help validate whether these computational distinctions manifest in real agent systems? If so, what kind of experiments would be most illustrative?

I am willing to increase the score if the above questions can be answered clearly, especially the Q1 and Q2. Q3 doesn't require running experiments but mainly what experiments can be designed around the theoretical claims.

---

> ### Author Response · Authors · 2025-11-16
>
> We thank Reviewer 3 for their time and constructive feedback.
> We will do our best to address all of their questions.
> We would like to answer the reviewer's questions in the order we feel is most effective for clarification.
>
>
> Q2:
> In practice, a reduction in a model's computability can manifest as execution-efficiency bottlenecks.
> Consider an agent handling a set of asynchronous subtasks, each consisting of three phases: (A) read input, (B) process, and (C) verify.
> A finite-context agent (Regular Language equivalent) must interleave these phases sequentially—`ABCABC...`—because it cannot remember prior unfinished work once its context window slides.
> This corresponds to a regular-language pattern, `(ABC)*`, which is easy to implement but scales poorly.
> In contrast, a retrieval-augmented agent (RQDP, Turing-complete) can instead batch operations as `A^N B^N C^N`,
> using its memory structure to defer verification until all inputs are read.
> This yields significant thread savings by pipelining asynchronous tasks.
> Since the `A^N B^N C^N` pattern requires memory beyond that of a finite-state machine, this is a win for models with higher computability.
>
> We also note that the preferred pattern itself depends on the external properties of the environment.
> This means that in a real-world scenario, the optimal inference pattern is not determined by the agent's inherent properties alone.
> Therefore, our paper analyzes, given a desired dependency graph by an external observer (such as one that implies an `A^N B^N C^N` execution),
> how efficiently each model of computation can simulate it.
> Our computation graph analysis in Section 6 discusses this.
> For example, we show that without retrieval mechanisms, even hierarchical agents (FQDPs) repeatedly solve identical subproblems instead of reusing results.
>
> Given the popularity of asynchronous programming and the use of computation graphs these days, we believe this argument perfectly answers the reviewer's question.
> This question has prompted us to add this explanation to the main paper to clarify the practical implications of our theory.
>
>
> Q3:
> Q3 follows directly from Q2: when an architecture lacks a memory mechanism that can preserve or retrieve intermediate results,
> it must re-compute shared dependencies, and its effective throughput degrades as predicted by our hierarchy.
> We believe that external performance metrics (e.g., wall-clock elapsed time) would directly reflect the computability distinctions we formalize.
> However, given the limited space, it is difficult to include a full and sufficiently varied experimental setup.
> This is why we instead provided the theoretical analysis in Section 6 based on computation graphs,
> which serves as a principled proxy for the empirical behavior the reviewer is asking about.
>
>
> Q1:
> Our hierarchy reflects patterns already present in many LM-based agents:
>
> Standard LMs with finite context align with our LM/FSM baseline, as their memory is limited to a fixed window.
>
> Systems that rely on sequence-of-thought or hierarchical task decomposition (e.g., ReAct (React: Synergizing reasoning and acting in language models), hierarchical RL)
> implicitly follow a QDP/FQDP-like structure, where reasoning unfolds by expanding and completing nested subgoals.
> Recent branching planners such as Tree-of-Thoughts exploration (Tree of thoughts: Deliberate problem solving with large language models)
> that incorporate external search mechanisms fall into the non-deterministic version of FQDP.
>
> Retrieval-augmented agents (e.g., RAG, memory-based tool use, long-term episodic memory agents) may match the RQDP tier only if their retrieval mechanism maintains state.
> Simple finite key–value RAG systems do not meet this requirement, as their queries depend solely on the current context.
> RAG without state reduces to FQDP or NFQDP depending on the agent's inference mechanism.
> We hypothesize that graph-structured retrieval systems such as HippoRAG (HippoRAG: Neurobiologically Inspired Long-Term Memory for Large Language Models)
> and LoopRAG (Enhancing Long Context Performance in LLMs Through Inner Loop Query Mechanism) may exhibit the necessary properties,
> though analyzing their exact equivalence would merit a dedicated study.
>
> We understand why the reviewer asked for this explicit mapping: in the paper, we provided citations but did not directly state the real-world equivalents of each automaton.
> With the additional rebuttal space, we have included brief sentences clarifying these mappings alongside the citations.
>
>
> --------
>
> This response was drafted by the author and edited paragraph-by-paragraph by an AI for readability only.

---

### Official Review · Reviewer_mdgD · 2025-11-01

**Soundness:** 2
**Presentation:** 3
**Contribution:** 3
**Rating:** 6
**Confidence:** 2

**Summary:**

This work introduces the Quest Graph framework to analyze computational capabilities of agentic systems with finite context windows. It links different architecture variants to distinct levels of the formal language hierarchy, clarifying the connection between agent design and expressive power.

**Strengths:**

- The paper connects agent architectures to models from automata theory (FSM, PDA, Turing machine), grounding abstract agentic reasoning in formal computability theory.
- The Quest Graph unifies reasoning, memory, and hierarchical task decomposition under one formal model, making it extensible to different types of agents (LLMs, hierarchical RL systems).
- The complexity analysis bridges theory and practice by showing the trade-off between computational expressiveness and execution efficiency.

**Weaknesses:**

- The framework remains purely theoretical, lacking experimental validation or benchmarking to demonstrate its applicability to real-world agentic systems or LLM-based agents.

**Questions:**

How might the Quest Graph framework be empirically validated through real-world LLM-based agents or reinforcement learning environments?

---

> ### Author Response · Authors · 2025-11-16
>
> We thank the reviewer for their time and valuable feedback.
>
> While we acknowledge that the lack of experiments may weaken the paper's position,
> we argue that the theoretical idea itself is significant and self-contained.
>
> Our core argument is that models with higher computability have more options to structure their reasoning,
> which in turn can lead to performance improvements (as discussed in our response to Reviewer 3).
> Therefore, performance-based measurements should reflect the underlying computability of the system.
>
> We note that some performance metrics may be defined by external factors, not just the inherent properties of the system.
> For example, in an asynchronous application, batching similar process calls (`AAA...BBB...CCC...`) is more beneficial than sequential calls (`ABCABC...`).
> Instead of performing experiments on various settings, we opted to show that given a desired inference pattern, how well each abstract automaton can simulate it.
> This is the reasoning for our analysis in Section 6.
>
> We have added a proper explanation of this logic to the main paper.
>
> ---
>
> Regardless of the final rating, we are committed to facilitating the reviewer's evaluation of this paper.
> If there is anything else that remains unclear, we would be more than happy to discuss it further to help the reviewer finalize their decision.
>
> ---
>
> This response was drafted by the author and edited paragraph-by-paragraph by an AI for readability only.

---

### Official Review · Reviewer_o9fc · 2025-11-01

**Soundness:** 3
**Presentation:** 2
**Contribution:** 2
**Rating:** 4
**Confidence:** 2

**Summary:**

In Computability of Agentic Systems, the authors provide a framework, principally derived from the notion of a novel Quest Graph (QG), to study the computational complexity of "agentic systems". The authors propose a series of Decision Processes that should represent typical agentic systems, along with common constraints, such as common reasoning patterns (QG -> GDP) captured through the action space (aka agent function space), limited context length (QDP -> FQDP), or breadth-first-search-like planning (FQDP -> NFQDP). Most importantly, the authors demonstrate that reference-augmented agentic systems ({FQDP, NFQDP} -> {RQDP, NRQDP}) are Turing complete, whereas typical non-reference-augmented agentic systems are not. The authors provide sound theoretical work, to the best of the reviewer's judgment.

**Strengths:**

The reviewer is not particularly familiar with the research domain of computational complexity for language models, except for the more popular work from that subdomain, such as "The Illusion of State in State-Space Models" (Merrill et al., 2024). However, they have a background in deep multi-agent RL and are familiar with MDP variants and agentic systems.


## Clarity
- For a theory-heavy paper, the reviewer could follow the paper quite well, which speaks to the logical structure in which the different decision processes and results were introduced and motivated.
- The design decisions to constrain the Quest Graph to better represent "agentic systems" are presented transparently.
- The paper cites previous work when necessary.

## Originality
- The attempt to formalise the computational complexity of "agentic systems" is worthwhile. The specific constraints on the Decision Processes seem original.

## Quality
- The proofs appear thorough. The reviewer could not identify any conclusion that was not properly motivated, at least in relation to the technical contributions.

## Significance
- Agentic systems using language models with finite context are showing great promise in different domains. A formal framework could help identify potential improvements in how such agentic systems should be constructed and facilitate the community's discussion of these decisions in a principled manner.

**Weaknesses:**

# Clarity


## Abstract


Overall, the abstract is fairly vague and could be improved. For example:

> "Theoretically, we demonstrate that these models form a hierarchy of computational power corresponding to key levels of the formal language hierarchy."

>  "We then analyze the practical efficiency of each model by simulating task dependencies in computation graphs, revealing that this
theoretical hierarchy translates to significant performance trade-offs".

(1) Please describe your findings directly, such as "We demonstrate that, given certain constraints, reference augmentation is necessary for an agentic system to be Turing complete" or "We show that reference-augmented systems have a better CG complexity than non-reference augmented decision processes.". Adapt accordingly.

(2) The reviewer disagrees that the paper is analysing "practical efficiency", as the computation graphs (MCGs) are still more of a theoretical construct than any decision process an agentic system might see in common agentic benchmarks. Under "practical efficiency", the reviewer would have expected a wall-clock time comparison on a popular agentic benchmark with the expected theoretical complexity, e.g., a reference augmented agentic system is faster on MLEBench than a non-reference augmented system.

## Main Text
### References
> Language models (LMs) lie at the core of these systems, serving as the primary decision-making module that processes language in a manner analogous to human cognition (Felin & Holweg (2024)). (line 28)

(3) Could you clarify how the Felin & Holweg citation motivates your statement of "analogous"? At least their abstract seems to disagree that any analogy is particularly meaningful when it comes to reasoning patterns and cognition:

> Scholars argue that artificial intelligence (AI) can generate genuine novelty and new knowledge and, in turn, that AI and computational models of cognition will replace human decision making under uncertainty. We disagree. We argue that AI’s data-based prediction is different from human theory-based causal logic and reasoning.

In general, I found the many references to human cognition distracting. For example, while the reviewer appreciates that the authors draw inspiration from cognitive science, stating that they were inspired by a theory of the hippocampus' mechanism (line 249), without providing further detail about what aspects of that theory inspired them, seems like unnecessary detail. Is it the fact that brains have long-term memory retrieval?

Also on line 328
> By keeping track of a location in the reference space, this mechanism functions analogously to cognitive maps in the hippocampus.

On what level does this analogy work? Where does this analogy break down? It appears to be a questionable scientific practice to make such broad statements, especially if the paper's principal contribution does not address human cognition and makes no effort to validate these statements empirically.

(4) The reviewer would prefer that these references be either solidified or left out completely, but is open to the author's opinions.

### Section 5
(5) The reviewer found Section 5 quite hard to follow; the explanation of the retrieve action could potentially be visualised, like in Figure 2.

(6) Similarly, could you please clarify the following paragraph (line 273)?
> Therefore, the reference-generating function must be allowed to vary per quest to ensure sufficient
capacity. Crucially, this design does not violate the agent’s stateless assumption. The reference graph
and its unbounded history are managed by the Quest Graph, while the agent function itself remains a
static, finite-context component unaware of this underlying complexity.

 It was not clear to the reviewer why the reference-generating function "must" be allowed to vary per quest.

# Significance and Originality
The reviewer found it challenging to grasp the significance and originality of the proposed framework fully. Most citations of contemporary work in the paper are methods applied to agentic systems. The citations are used to motivate the paper's design decisions. There are very few citations for related work that compare and contrast this paper with their academic "siblings". It's quickly mentioned that other "unconventional models" exist in the literature, but no effort was made to explain how or if they differ. Furthermore, as the authors point out, formally, a QDP is a variant of a POMDP. Sequence models, such as RNNs, and even feedforward neural networks, have been utilised in POMDPs for a considerable time.

(7) Could the authors clarify if there are any comparable frameworks for complexity analysis on POMDPs? What makes this analysis unique to LMs? Do they consider state-space models and LSTMs also as LMs, or is this work mostly focused on transformer-based architectures?

Furthermore, the authors state that,
> The standard model of an LM treats the decision-making module and the context buffer as a single,
monolithic entity. We propose decoupling these components, presenting the reasoning module as a
pure function that operates on an external memory structure. This separation offers both practical and
theoretical advantages. On a practical level, it aligns with modern, stateless architectures recognized
for their scalability and testability (Fielding (2000)).

(8) Could the authors please clarify what that means for the transferability of their framework to practical usage of agentic systems? Moreover, Fielding (2000) appears to discuss stateless architectures of network-based systems in the context of the World Wide Web. Could you clarify the connection to LM-based agentic systems?

Moreover, without any empirical results, it's hard to agree that the design choices for action spaces and retrieval augmentation are genuinely meaningful. While the proofs appear sound, given the action space design, it remains unclear whether this framework is predictive of the actual computational complexity of LM-based agentic systems in the end. The design choices were mostly motivated by citing some related method, but it's unclear if that motivation is sufficient.

(9) Would the authors be willing to include empirical results, such as comparing different agentic systems (matching their proposed decision processes) on a single, simple agentic, even if contrived, benchmark? While probably out of scope, work like [1] compares AIDE and AIRA as agentic systems with different node selection techniques on MLEBench. Would it be feasible to make predictions about the performance of these systems with your framework?

(10) Alternatively to (9), could the authors further motivate how their choices for the action space are necessary and sufficient to represent agentic systems? While intuitive, it is hard to grasp why, e.g., the agent function in Section 3.1 only has the 3 subactions


[1] AI Research Agents for Machine Learning: Search, Exploration, and Generalization in MLE-bench, Toledo et al. 2025,

**Questions:**

The reviewer listed all questions in the Weakness section, enumerated. The reviewer is happy to engage in a follow-up discussion on these questions and clarify any points as necessary.

If the authors could solidify the originality and significance of the contribution, and compare and contrast it with other related work, if it exists, the reviewer would be happy to consider changing their score. Any similar work that was accepted at previous conferences would also help the reviewer calibrate their decision.

---

> ### Author Response · Authors · 2025-11-16
>
> We greatly appreciate the reviewer for providing this constructive feedback,
> and we will do our best to address all points. We must keep our responses brief due to space limits.
> If any of our answers seem abrupt or convey a mistaken sense of disrespect, we assure the reviewer this is not our intention.
> It would only be a result of our command of the language, and we apologize in advance.
>
> To ease the narrative, we would like to begin by addressing Significance and Originality.
>
> --------
>
> Significance and Originality
>
> The goal of this paper is to "abstract" agentic systems and to analyze their computability patterns and efficiency.
> We recognize that the main text was lacking in direct comparisons to existing models (a point also raised by Reviewer 3).
> We have made the proper edits to the main text to include these direct comparisons.
>
> Q7:
> To the best of our knowledge, "Rethinking Transformers in Solving POMDPs" directly corresponds to the first three parts of this question.
>
> Transformers are TC0, which is believed to be a sub-class of Regular Languages (RE) (Hahn 2020).
> RNN and LSTM are on par with RE.
> POMDPs are believed to be on a tier higher than RE (thanks to their unbounded history).
> The potential of a POMDP to expand depends on its action space and can achieve Turing completeness.
> In this work, however, we only focus on a specific subset of POMDP where the hierarchy action is added.
> We argue that if a POMDP is augmented with a hierarchy function in the way we present, it becomes a Context-Free Language.
> We therefore consider the computability of POMDP as a whole to be not directly related to our work.
>
> Also, we did not mention Transformers as LMs (though the text implies that, for which we apologize).
> Rather, we present an LM as an abstract auto-regressive machine that can be implemented by underlying choices: RNN, LSTM, or Transformer.
> We show that this abstraction is proven to be RE.
>
> But the reviewer is right. Our choice of citation does not reflect this well; we have made modifications to the main text to reflect this.
>
>
> Q8:
> The practical way people use LMs these days is via headless services provided by third parties, which are a form of web service.
> Our use of "stateless" means the service provider does not keep track of the caller system's state.
> Instead, everything has to be passed as arguments to the service at each call, just like our abstract agent function.
> This behavior is reflected in serverless systems (Architectural styles and the design of network-based software architectures) and in the choice of abstract machines we presented.
>
>
> Q9:
> Due to limited time and resources, we were unable to conduct such experiments for this submission.
> We will, of course, accept any decision based on the merits of the existing (and modified) main text.
> However, this question is similar to those raised by Reviewers 2 and 3 regarding experimental validation.
> Namely, they ask what practical aspects our analysis ties to and how they could be measured.
>
> As an example, we argue that thanks to the widespread use of asynchronous patterns,
> there is a clear efficiency advantage in arranging service calls as AAA..BBB..CCC.. instead of ABCABCABC...,
> especially when B is an asynchronous process that should be pipelined.
> The interleaved pattern (ABCABC...) can be handled by a Regular Language machine,
> but the batched pattern (AAA..BBB..CCC..) requires Context-Sensitive capabilities.
> This means computability directly impacts system efficiency and could be measured in an experiment that captures wall-clock time.
> (We would refer the reviewer to our discussion for Reviewer 3 on this topic.)
>
> This example illustrates our core argument: an external metric (like minimizing wait time for asynchronous tasks) defines a preferred execution pattern.
> The system's underlying computability (Regular vs. Context-Sensitive) then dictates whether it can efficiently achieve that pattern.
> This is precisely what we analyze in Section 6.
> Instead of performing one empirical experiment, we ask a more general question:
> given a desired dependency pattern, how efficiently can each of our abstract models simulate it?
> This serves as a theoretical guarantee of these efficiency trade-offs.
>
> Due to space and time limitations, we could not include a full empirical analysis in our paper.
> But we believe that the points we made here—the link between abstract machines and their computability—are self-contained enough and ready to elicit further analysis.
> This is why we decided to publish this work in its current state.

---

> ### Author Response · Authors · 2025-11-16
>
> Q10:
> We can link our action space choices to existing systems.
> For example, in the ReAct framework, an agent's action is to select proper tools (or sub-agents) step-by-step.
> This corresponds to our "Discover" node action.
> When the selected tool is finished, it returns the context to the parent agent, which corresponds to our "Respond" action.
> The "Stop" action in ReAct is a variant of this "Respond" action, but for the root node.
> Furthermore, the "thought" action in ReAct can be implemented by a sequence of "Discover" and immediate "Respond" actions.
>
> Our choice of action space was selected as the minimal set of actions required for an agentic system to sufficiently operate.
>
>
> Q5 and Q6:
> Due to space limits, we would like to offer a detailed explanation here.
> If this clarifies our answers, we will incorporate a condensed version into the main text.
>
> The core of our retrieval mechanism (Q5) is the "reference-generating function".
> For any retrieval system to work, it must have a way to distinguish one record of knowledge from another, which we call a "tag" or "reference".
> This function is what produces such a tag.
> Every time a new node is attached to the Quest Graph, this function assigns it a unique tag.
> When the agent uses the "retrieve" action, a new node is created.
> However, unlike the standard "discover" action, the Quest Graph checks this new node's tag against the pool of all past nodes.
> If it finds a match, it copies the response content from the corresponding past node into the new node.
> This allows old knowledge to be retrieved and reused in the current context.
>
> This design directly explains Q6: why the reference-generating function must be unbounded and external to the agent.
> The agent function itself is finite (it operates with finite parameters).
> If the Quest Graph runs indefinitely, the number of nodes created will eventually exceed the total number of possible configurations of the finite agent.
> To ensure every node can still be uniquely distinguished, the tags must come from an unbounded space.
>
> This is precisely why the reference and tag system is "hidden" from the agent and managed by the Quest Graph.
> The agent function is only ever presented with its finite set of goals and responses.
> This separation allows the agent to remain a simple, finite-context component while the system as a whole can access and manage an unbounded history.

---

> ### Author Response · Authors · 2025-11-16
>
> Q1:
> We agree with the reviewer. Thanks to the space extension, we will make the appropriate changes to the abstract and conclusion as suggested.
>
>
> Q2:
> We agree with the reviewer; our original choice of "practical efficiency" was misleading.
> We have changed this term to "theoretical efficiency" throughout the paper.
> This error was due to our initial intention for Section 6 to serve as a substitute for empirical analysis,
> and that impression unfortunately carried over into our final choice of words.
>
>
> Q3:
> We apologize for this error. It occurred during an attempt to reduce the line count.
> Our original intention was to use Niu et al. (2024) (Section 3A) as a supporting citation and Felin & Holweg (2024) as a counter-example to our point of view.
> Unfortunately, we inadvertently mixed up the citations during editing.
> In response to this and Q4, we have opted to remove these references to prevent further confusion.
>
>
> Q4:
> We would like to explain the connection, but we believe it cannot be adequately justified within the space limitations. Therefore, we agree to remove it.

---

> ### Author Response · Authors · 2025-11-16
>
> Perhaps our primary goal in publishing this work is to provide a framework that allows designers of agentic systems a scaffold for reasoning about their system's complexity.
>
> This relates directly to our argument for Q9.
> Computability is not just a theoretical curiosity;
> it also defines a system's flexibility to handle process dependencies, which in turn reflects efficiency.
> However, within the scope of one paper, we could not fairly conduct an empirical analysis of actual, rapidly-evolving systems.
> We therefore chose to align with our expertise and carefully design abstract machines based on these systems.
> System designers can reduce their own systems to our minimal set of actions and benefit from the theoretical connections we have provided.
>
> --------
>
> This response was drafted by the author and edited paragraph-by-paragraph by an AI for readability only.

---

> > ### Comment · Reviewer_o9fc · 2025-11-18
> >
> > The reviewer acknowledges the authors' responses. The questions were sufficiently clarified, and the necessary changes were made. The reviewer thus suggests accepting the paper, given the solid theoretical contributions.

---

### Author Response · Authors · 2025-11-22
**Summary of changes**

We sincerely appreciate the reviewers' time and effort in providing constructive feedback.
We also thank the Area Chair for the opportunity to share and discuss our research.

Below is a summary of the key changes we have made to the manuscript:

### Motivation and Real-World Anchors
- We have added explicit mappings for each abstract machine to its existing real-world equivalent. This addresses the concerns raised by Reviewer 3 (as well as Reviewers 1 and 2) regarding the practical grounding of our framework.
- Furthermore, we have revised the introduction to include the argument that computability directly affects efficiency, creating a more compelling opening. We hope this clarifies the significance of our contribution.

### Presentation and Text Quality
- **Terminology:** We adopted more precise terms, specifically changing "practical efficiency" to "theoretical efficiency" throughout the text.
- **Abstract & Conclusion:** We improved both the Abstract and Conclusion to explicitly state our key findings regarding the computational hierarchy and efficiency trade-offs.
- **Table 1:** We moved Table 1 to the end of the introduction to serve as an illustrative overview early in the paper.
- **Citations:** We removed references regarding inspiration from cognitive science to improve focus.
- **Formatting:** We adjusted figure positions to improve readability and reduce page flipping.
- **Section 5:** As promised to Reviewer 1, we have rewritten the text in Section 5 (RQDP) to clarify the retrieval mechanism. We hope this will help the audience understand the content better.
- **Section 5:** We refined the literature review on extensions of PDAs to explicitly specify the use of a standard logarithmic work tape and updated the relevant references.
- **Section 6:** We refined the statement regarding the limitations of LLMs in simulating computation graphs.
- **Section 6:** We added a justification for the logarithmic access time assumption used in our complexity analysis.

We would be delighted if our revisions and replies assist the reviewers in their final assessment. If anything remains unclear, please do not hesitate to ask; we will do our best to respond. Finally, if we have missed any specific points, we apologize in advance and kindly ask that you re-raise them.

---

### Meta-Review · Area_Chair_kkaZ · 2026-01-02

**Summary:**

### paper summary

This work introduces the Quest Graph framework to analyze computational capabilities of agentic systems with finite context windows. It links different architecture variants to distinct levels of the formal language hierarchy, clarifying the connection between agent design and expressive power.

### reviewer feedback

Reviewers generally find the theoretical framing and hierarchy claims well-motivated and useful as a shared language for agentic-system design.
The Section 6-style complexity story is viewed as a meaningful bridge from computability to performance trade-offs (step complexity / exponential vs polynomial).
However, multiple reviewers emphasize that without at least small controlled experiments, it’s hard to judge whether the separations meaningfully manifest in realistic agent stacks.  Even after added mappings, at least one reviewer still felt the paper under-delivers on where current systems fail and what concrete design guidance follows. Moreover, modeling assumptions could mislead readers about wall-clock implications and leave open how stochasticity changes the hierarchy.

### AC comments

the paper still provides no empirical evidence that the proposed complexity separations are observable in real agent pipelines, and the practical relevance argument remains largely narrative and proxy-based. Given the split reviews and the fact that the strongest concern is not actually resolved, I do not think the current version meets the bar.

**Reviewer Concerns:**

1. Resolved

* Misleading practical efficiency phrasing .
* Distracting cognition-inspired citations/claims.
* RQDP retrieval explanation
* Mapping abstractions to real systems


2. Unresolved

* No empirical validation remains the dominant unresolved issue: several reviewers explicitly asked for even smallexperiments or case studies, and the authors did not provide them.
* Actionable guidance still not fully convincing: even with added mappings, at least one reviewer remains unconvinced that the framework yields concrete, testable design recommendations for today’s agent stacks.
* authors clarify they analyze step complexity, but the gap to real stochastic agents and practical runtime remains a material limitation.

**Reviewer Scores:**

* o9fc indicates the clarifications were sufficient and flips to acceptance.

* mdgD's main concern is still the lack of experiments; no new empirical evidence was added. So they will keep scores.

* Hqpp could move slightly with improved mapping/writing, but the absence of validation/actionable evidence likely keeps them below threshold.

* 6Fk5 score likely unchanged.

---

### Decision · Program_Chairs · 2026-01-26

Reject